# Reactive astrocytic S1P3 signaling modulates the blood–tumor barrier in brain metastases

Brunilde Gril [1], Anurag N. Paranjape [1], Stephan Woditschka[1,17], Emily Hua[1], Emma L. Dolan[1], Jeffrey Hanson[2], Xiaolin Wu[3], Wojciech Kloc[4,5], Ewa Izycka-Swieszewska [6,7], Renata Duchnowska [8], Rafał Pęksa [9], Wojciech Biernat[9], Jacek Jassem[10], Naema Nayyar[11], Priscilla K. Brastianos[11], O. Morgan Hall[12], Cody J. Peer[12], William D. Figg[12], Gary T. Pauly [13], Christina Robinson[14], Simone Difilippantonio[14], Emilie Bialecki[15], Philippe Metellus[15,16], Joel P. Schneider[13] & Patricia S. Steeg[1]

Brain metastases are devastating complications of cancer. The blood–brain barrier (BBB), which protects the normal brain, morphs into an inadequately characterized blood–tumor barrier (BTB) when brain metastases form, and is surrounded by a neuroinflammatory response. These structures contribute to poor therapeutic efficacy by limiting drug uptake. Here, we report that experimental breast cancer brain metastases of low- and high permeability to a dextran dye exhibit distinct microenvironmental gene expression patterns. Astrocytic sphingosine-1 phosphate receptor 3 (S1P3) is upregulated in the neuroinflammatory response of the highly permeable lesions, and is expressed in patients' brain metastases. S1P3 inhibition functionally tightens the BTB in vitro and in vivo. S1P3 mediates its effects on BTB permeability through astrocytic secretion of IL-6 and CCL2, which relaxes endothelial cell adhesion. Tumor cell overexpression of S1P3 mimics this pathway, enhancing IL-6 and CCL-2 production and elevating BTB permeability. In conclusion, neuroinflammatory astrocytic S1P3 modulates BTB permeability.

[1] Women's Malignancies Branch, CCR, NCI, Bethesda 20892 MD, USA. [2] Laboratory of Pathology, CCR, NCI, Bethesda 20892 MD, USA. [3] Genomics Laboratory, Frederick National Laboratory for Cancer Research, Frederick 21702 MD, USA. [4] Department of Neurology & Neurosurgery, Varmia & Masuria University, Olsztyn 10-719, Poland. [5] Department of Neurosurgery, Copernicus Hospital Gdańsk, Gdańsk 80-803, Poland. [6] Department of Pathology & Neuropathology, Medical University of Gdańsk, Gdańsk 80-210, Poland. [7] Department of Pathomorphology, Copernicus Hospital Gdańsk, Gdańsk 80-803, Poland. [8] Department of Oncology, Military Institute of Medicine, Warsaw 04-141, Poland. [9] Department of Pathology, Medical University of Gdańsk, 7 Dębinki St, 80-211 Gdańsk, Poland. [10] Department of Oncology and Radiotherapy, Medical University of Gdańsk, Gdańsk 80-211, Poland. [11] Division of Neuro-Oncology, Massachusetts General Hospital Cancer Center, Harvard Medical School, Boston 02114 MA, USA. [12] Genitourinary Malignancies Branch, CCR, NCI, Bethesda 20892 MD, USA. [13] Chemical Biology Laboratory, CCR, NCI, Frederick 21702 MD, USA. [14] Laboratory Animal Sciences Program, Frederick National Laboratory for Cancer Research, Frederick 21702 MD, USA. [15] Département de Neurochirurgie, Hôpital Privé Clairval, Ramsay Général de Santé, Marseille 13009, France. [16] Institut de Neurophysiopathologie—UMR 7051, Aix-Marseille Université, Marseille 13344, France. [17] Present address: Department of Biology and Marine Biology, University of North Carolina at Wilmington, 601 South College Road, Wilmington, NC 28403, USA. These authors contributed equally: Brunilde Gril, Anurag N. Paranjape. Correspondence and requests for materials should be addressed to B.G. (email: grilbrun@mail.nih.gov) or to P.S.S. (email: steegp@mail.nih.gov)

The blood–brain barrier (BBB) limits brain uptake of most compounds. The BBB consists of endothelial cells with continuous tight junctions and efflux pumps, endothelial and parenchymal (astrocytic) basement membranes, pericytes, and the feet of astrocytes. Tumors in the brain, whether primary or metastatic, alter the BBB. This remodeled structure, the blood–tumor barrier (BTB), is surrounded by a neuroinflammatory response.

Breast cancer is the second leading source of brain metastases, predominantly in patients with metastatic disease that is either HER2+ or triple-negative (estrogen and progesterone receptors-negative, HER2 normal)[1,2]. Brain metastases and the consequences of their treatment are particularly devastating in terms of neurocognitive complications. Increasingly, they contribute to patient deaths.

The role of BTB permeability in the treatment of brain metastases is complex. Clinical brain metastases are diagnosed using gadolinium uptake, indicating some BTB permeability. Hematogenously derived brain metastases in mice demonstrated heterogeneous uptake of markers and drugs at levels higher than normal brain, but with peak levels approximately a log less than systemic metastases[3–8]. Capecitabine and lapatinib levels in surgically resected breast cancer brain metastases were shown to have a large variability of brain metastases-to-serum ratios, confirming heterogeneous drug uptake through the human BTB[9]. Thus, the BTB may be only partially and heterogeneously permeable, with potentially profound consequences for drug efficacy. A pharmacokinetic study in mice bearing brain metastases of breast cancer used tumor cell response as a functional measure of lesion permeability: using paclitaxel, only about 10% of the metastases with the highest uptake of drug (50-fold above BBB) demonstrated apoptosis in vivo[4]. Similar data were reported for vinorelbine[10]. The low drug uptake in preclinical models is consistent with data from multiple clinical trials using drugs that were effective for systemic metastatic disease, but showed no significant activity against brain metastases[11–16].

Our goal is to understand the permeability of the BTB to improve drug uptake in brain metastases. We recently characterized the BTB in three experimental models of brain metastasis of breast cancer, triple-negative MDA-MB-231-BR6[17] (231-BR), HER2+ JIMT-1-BR3[18] (JIMT-1-BR), and HER2+ SUM190-BR3[19] (SUM190-BR). Consistent changes were found in the transition from a BBB to a BTB, including the endothelial, neuroinflammatory, pericyte, basement membrane, and astrocytic components. When brain metastases of low- vs. high permeability to the fluorescent dye 3 kDa Texas Red Dextran (TRD) were compared, fewer BTB changes were observed: in three model systems, highly permeable metastases were characterized by a loss of laminin α2 in the astrocytic basement membrane, a loss of a CD13+ subpopulation of pericytes, and a gain in a desmin+ subpopulation of pericytes[19]. Herein, we hypothesize that additional molecular alterations correlate with low- vs. high permeability metastases, undetectable by an IF screen of known BBB components. For instance, while the number of GFAP+ astrocytes did not differ between low- and high permeability lesions, these cells may use diverse pathways. To test this hypothesis, laser capture microdissection (LCM) of low- and high permeability brain metastases was performed and gene expression profiled. The sphingosine 1-phosphate receptor 3 (S1P3) was differentially expressed in the astrocytic neuroinflammatory response of low- and high permeability metastases.

We report that S1P3 is overexpressed by astrocytes in the neuroinflammatory response from highly permeable brain metastases, as compared to less permeable metastases in the same mouse brain, in four independent model systems. S1P3 is detectable in reactive astrocytes from patients' brain metastasis samples. Functional studies in vitro and in vivo demonstrate reduced BTB permeability with S1P3 inhibition, mediated by multiple astrocyte-derived cytokines and resultant alterations in the BTB endothelia. Overexpression of S1P3 by brain-metastasis cells recapitulate this pathway, elevating cytokine production and increasing BTB permeability. The data provide a proof of principle that BTB permeability can be modulated, herein via the S1P3 astrocytic neuroinflammatory response.

## Results

**Transcriptome of low- and high permeability brain metastases.** To study the BTB at the molecular level, gene expression was compared between LCM metastases of relatively low and high permeability to TRD within the same mouse brain (Fig. 1a). TRD was used as a marker of paracellular permeability, as its uptake in experimental brain metastases correlated directly with that of paclitaxel, vinorelbine, doxorubicin, and lapatinib[4,10,20]. The gene expression patterns of the human 231-BR tumor cells and mouse tumor microenvironment were captured using human and mouse microarrays, respectively. Of the 39,000 mouse transcripts and 47,000 human transcripts, 2748 murine and 1097 human were differentially expressed ($P < 0.05$, $t$-test) (Supplementary Data 1 and 2). Figure 1b, c presents heatmaps of differentially expressed genes from the mouse and human microarrays, respectively. Profound gene expression differences were most apparent in the mouse data, suggesting the importance of brain microenvironmental regulatory pathways. Highly permeable metastases were characterized by downregulation of murine ephrin receptors *Epha3* and *Epha5*, gamma-aminobutyric acid (GABA) A receptor (*Gabra*), subunits α1, α2, and γ1, the tight junction component cingulin (*Cgn*), activated leukocyte cell adhesion molecule (*Alcam*), monoamine oxidase involved in a metabolic barrier (*Maob*), Glutamate receptor, metabotropic 8 (Grm8), and glutamate receptor ionotropic, NMDA3A (Grin3a). Moreover, highly permeable metastases overexpressed sphingosine-1-phosphate receptor 3 (S1pr3), a microglial protein (*Cd68*), the chemokine ligand 5 (*Ccl5*), and granulocytic/lymphocyte homing molecules (*Lrg1, Stab1*) (Fig. 1b). Ingenuity® pathway analyses identified glutamate, GABA, G-protein-coupled receptors, ephrin receptor, and axonal guidance pathways (Supplementary Fig. 1). Low permeability metastases were similar in gene expression to the BBB from normal mice. The human microarray analysis yielded fewer significantly altered pathways. Differentially expressed human genes included overexpression of nicotinamide N-methyltransferase (*NNMT*) and C-type lectin domain family 2, member B (*CLEC2B*) in the highly permeable lesions and overexpression of synovial sarcoma, X breakpoint 1 (*SSX1*), Granzyme 1 (*GZMA*), gap junction protein, beta 2 (*GJB2*), cell adhesion molecule 1 (*CADM1*), and AF4/FMR2 family member (*AFF3*) in the less permeable lesions (Fig. 1c). Most of these genes are not traditional parts of the BBB.

Differential gene expression trends for several genes were validated at the protein level using quantitative immunofluorescence (IF) (Fig. 2). An independent set of brains from mice carrying 231-BR metastases, injected with TRD and then perfused at necropsy, were used. EphA5, GABA A receptor subunits, and S1P3 were expressed by subpopulation of activated astrocytes in the neuroinflammatory response, with EphA5 also expressed by neurons. Both herein and in a previous study[19], the density of activated astrocytes was similar in poorly and highly permeable lesions (Supplementary Fig. 2), although their morphologies varied (Supplementary Fig. 3). For the GABA A receptor, GABA A γ2 subunit mRNA was 1.6–1.7-fold downregulated in highly permeable as compared to less permeable lesions in the same brain using two probe sets ($P < 0.05$, $t$-test). GABA A γ2 staining

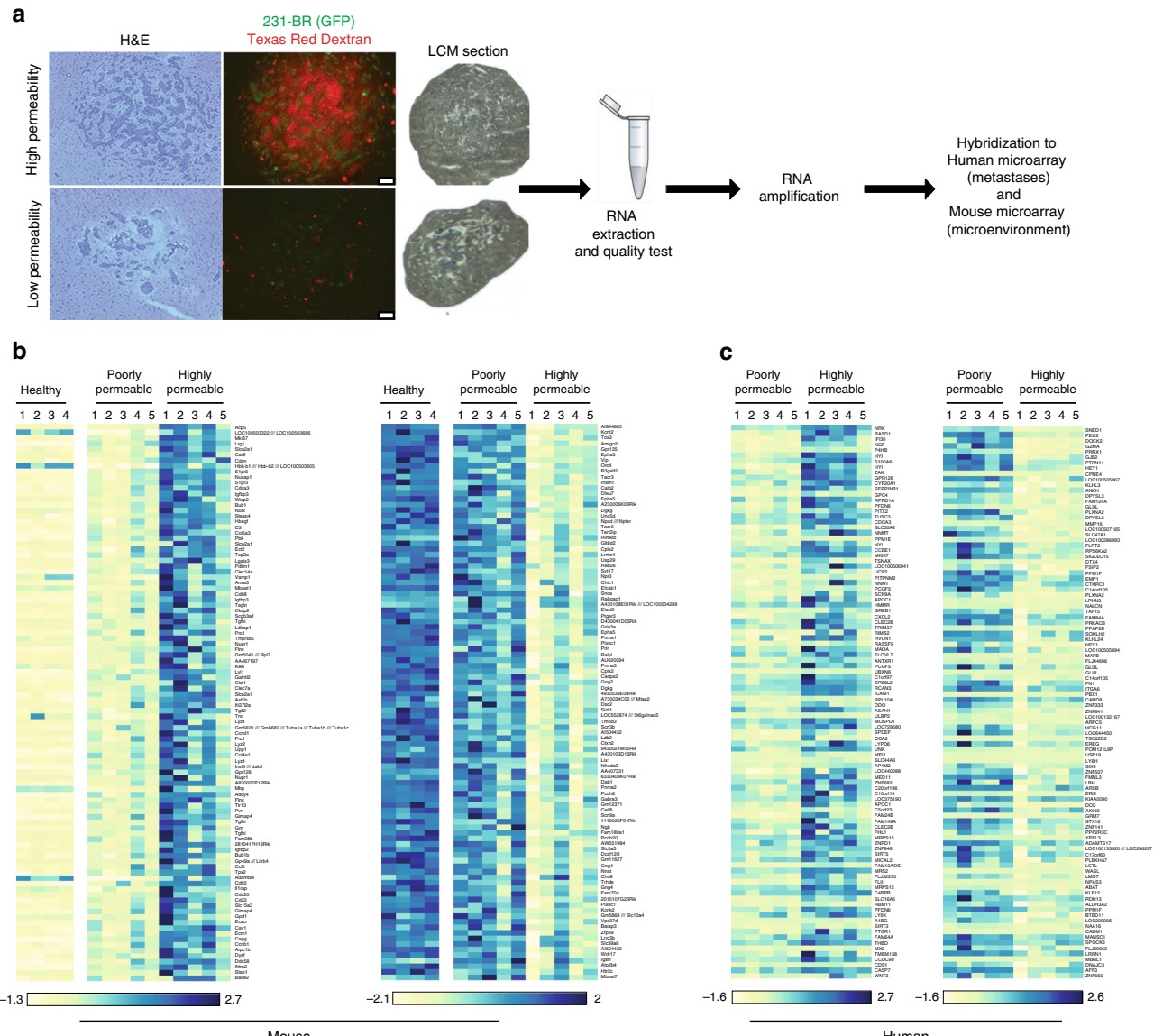

**Fig. 1** Differential gene expression in poorly and highly permeable experimental brain metastases of breast cancer. Mice harboring GFP+ brain metastases of MDA-MB-231-BR (231-BR) triple-negative breast cancer cells were administered 3 kDa Texas Red Dextran (TRD) as an indicator of paracellular permeability, and perfused at necropsy to remove TRD from blood vessels. Highly permeable and poorly permeable metastases from a single mouse brain were collected by laser capture microdissection, and microarray gene expression analysis performed. Both metastatic cells and the immediate microenvironment were captured for five mice. **a** Example of poorly and highly permeable metastases from a single mouse brain and a schematic of the procedure. A section showing TRD exudation (red) and the adjacent H&E stained section are shown (scale bar = 100 μm). **b** Differentially expressed murine (brain microenvironment) genes between poorly and highly permeable metastases. Normal brains from healthy mice were included for comparison. **c** Differentially expressed human (metastatic cells) genes between poorly- and highly-permeable metastases. Numbers above indicate mouse number

normalized to total activated astrocytes (GABA A γ2 /GFAP) was downregulated in the highly permeable as compared to poorly permeable lesions in the same brain ($P = 0.023$, Wilcoxon matched-pairs signed rank test) (Fig. 2a). Similar results were obtained for GABA Aα1 (Fig. 2b). The ephrin receptor EphA5 was downregulated at the mRNA level in highly permeable metastases by 2.1–2.9-fold using three probe sets ($P < 0.05$, t-test). EphA5 staining was significantly reduced in the highly permeable lesions of all but one of 11 examined brains ($P = 0.002$, Wilcoxon matched-pairs signed rank test) (Fig. 2c). Other genes could not be confirmed at the protein level due to a lack of staining specificity on frozen tissues. The data confirm several of the differential gene expression trends at the protein level. They support the hypothesis that, even though astrocyte densities were

comparable in highly permeable and less permeable brain metastases, gene/protein expression trends in these cells may be distinct.

**Overexpression of S1P3 in highly permeable metastases**. The sphingosine-1 phosphate (S1P) receptor 3 gene *(S1pr3)* was selected from the mouse microarray data for further validation based on its 2.4–2.5-fold overexpression in highly permeable as compared to poorly permeable lesions in the same brain, using two probe sets ($P = 0.025$, t-test). S1P3 is one of five members of the S1P G-protein-coupled receptor family, involved in anti-apoptotic, proliferative, and inflammatory signaling, primarily reported in the immune system and endothelia[21–28]. In

the normal brain, S1P promotes developmental and survival roles, regulating neurotrophic factors, neurotransmitter release, microglia and astrocyte activation, autophagy, and anti-oxidant and anti-apoptotic responses[22]. In the experimental metastasis model used herein, none of the other S1P receptor family members were differentially expressed. S1P3 signals through pathways both common and distinct from its family members including $G_i$ to

cAMP/PI3K/ERK, $G_{12/13}$ to RHO, and $G_q$ to phospholipase C (PLC). The normal BBB exhibits S1P3 staining on astrocytes and endothelial cells[24,29], and enhanced astrocytic S1P3 expression occurs in multiple sclerosis (MS)[21,30–32]. IF co-staining studies demonstrated S1P3 expression by a subpopulation of GFAP+ astrocytes in the neuroinflammatory response surrounding metastases (Fig. 3a and Supplementary Fig. 4 using an

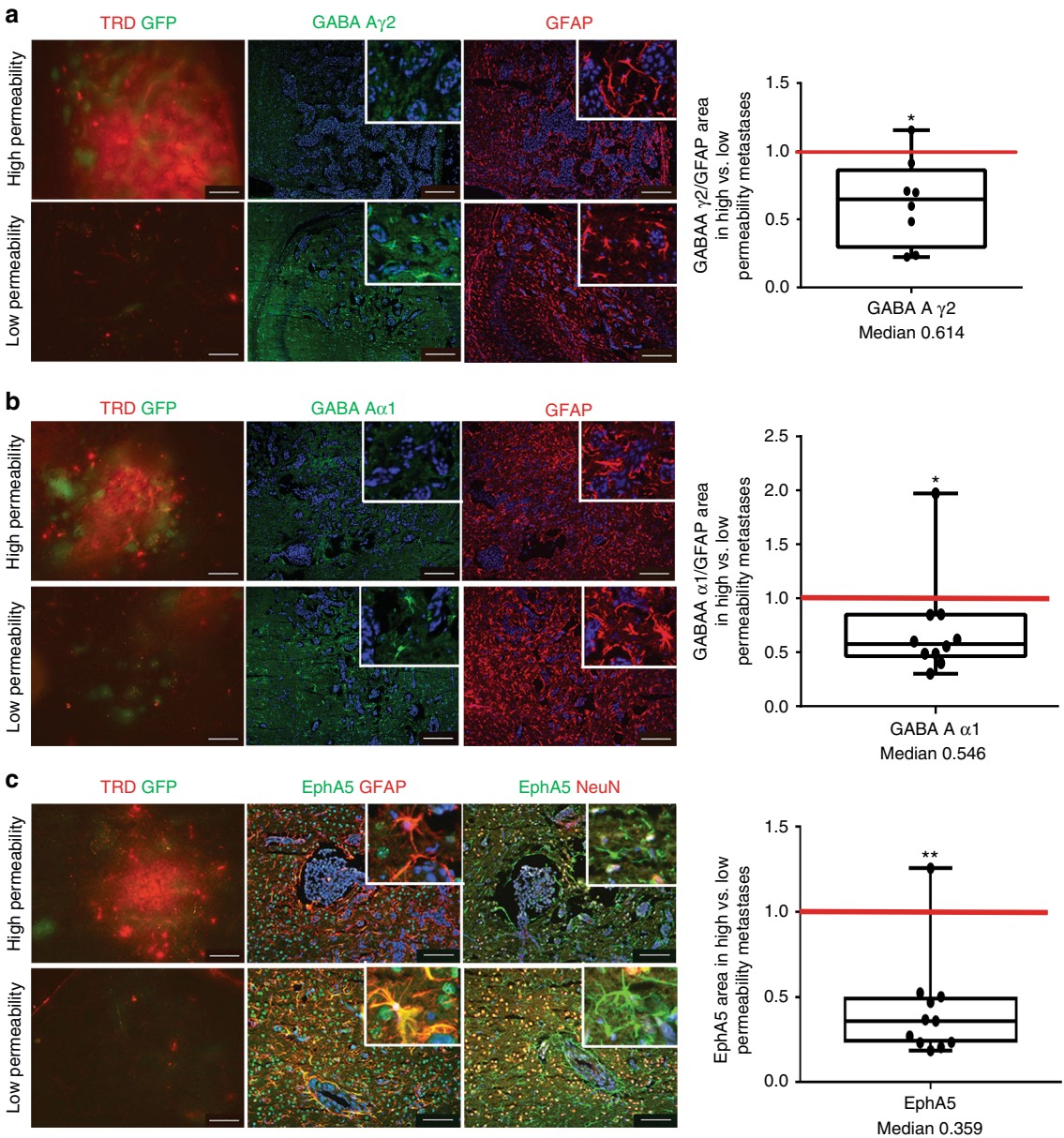

**Fig. 2** Validation of three differentially expressed microenvironmental genes at the protein level. **a–c** Using an independent cohort of mice harboring GFP+ 231-BR brain metastases, metastases were identified in H&E stained step sections, and those distinctly low and high in permeability to TRD identified by red fluorescence exudation. Sections in between the H&E and TRD markers were used for immunofluorescence (IF) staining. Examples of TRD exudation with IF staining in the adjacent sections for the protein of interest and the cell source are shown. For each metastasis of known permeability, the area of IF signal for a protein was normalized to the area of staining for the brain cell type(s) in which it was expressed; the ratio of staining intensity in all highly permeable/low permeable metastases in each mouse brain is shown as a single dot on the graph to the right. The red line at unity would indicate no trend in highly/poorly permeable metastases. **a** Representative staining for GABA Aγ2 (green), and GFAP (red) as an activated astrocytic marker for the surrounding neuroinflammatory response; DAPI stained nuclei were blue in all sections. Insets: image magnification. GABA Aγ2 staining intensity was normalized to GFAP intensity, and was compiled as a ratio of highly/poorly permeable metastases in a single mouse brain, represented on the graph as a dot. Median = 0.614 (n = 8). **b** A second differentially expressed GABA A receptor protein, Aα1. Median = 0.546 (n = 10). **c** Similar staining for Ephrin receptor EphA5, except that expression was documented in both GFAP+ astrocytes and NeuN+ neurons. EphA5 in highly/poorly permeable metastases, median = 0.359 (n = 11). From (**a**) to (**c**): graphs represent the median, the interquartile range (box) and the min/max values (error bars). Wilcoxon matched-pairs signed rank test, two-tailed, *P < 0.05, **P < 0.01 (scale bar = 100 μm)

independent antibody); no detectable staining of endothelia was found (Supplementary Fig. 5). S1P3+ astrocytes were absent from uninvolved brain (Supplementary Fig. 6). S1P3 staining normalized to total activated astrocytes (S1P3/GFAP fluorescence) was quantified in highly and poorly permeable brain metastases from the same mouse brain in four model systems: an independent set of 231-BR, the HER2+ JIMT-1-BR[18] and SUM190-BR[19], and the 4T1-BR immunocompetent models (Fig. 3b–e). GFAP+ astrocyte

staining was similar in highly and poorly permeable brain metastases (Supplementary Fig. 2)[19]. In the 231-BR model, the median staining intensity of S1P3/GFAP was elevated 1.73-fold ($P = 0.052$, Wilcoxon matched-pairs signed rank test, two-tailed) in highly permeable compared to poorly permeable metastases (Fig. 3b), indicating a trend in S1P3 overexpression in astrocytes surrounding the more permeable lesions. The JIMT-1-BR and SUM190-BR brain metastasis model systems confirmed this

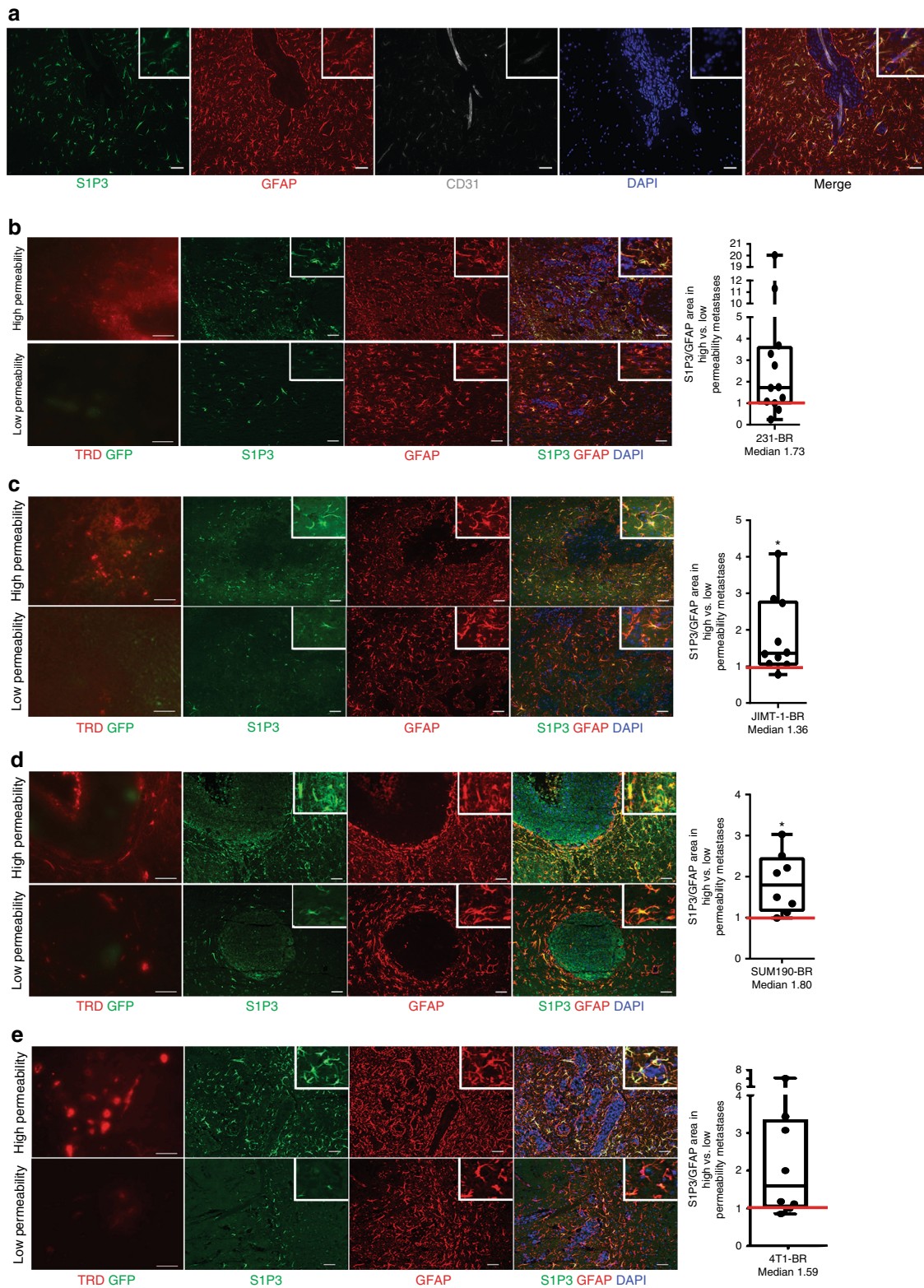

trend, with median ratios of S1P3/GFAP in highly permeable/less permeable lesions of 1.36 ($P = 0.014$) and 1.80 ($P = 0.016$), respectively (Fig. 3c, d). In the 4T1-BR model, a trend of S1P3/GFAP overexpression in highly permeable lesions was observed with a median ratio of 1.59 ($P = 0.055$) (Fig. 3e). Thus, overexpression of S1P3 in the astrocytic neuroinflammatory response surrounding the most permeable brain metastases was evident in four independent model systems.

**Human craniotomy tissues**. We asked if astrocytes in the neuroinflammatory response of human brain metastases also expressed S1P3. IF co-staining of S1P3 and GFAP was optimized on fresh frozen tissues, which limited the number of brain metastases samples from lung or breast cancer patients available to 19 (Fig. 4). Virtually all of the cells that stained for S1P3 were morphologically astrocytes and GFAP+, but not all GFAP+ astrocytes were S1P3+. The percentage of GFAP+ astrocytes expressing S1P3 was highly variable, from 9.6 to 81% (Supplementary Data 3). S1P3 staining of endothelial cells was not detected. An independent cohort of 31 paraffin-embedded human brain metastasis specimens was immunostained for S1P3. A majority (68%) of the specimens presented S1P3 staining in astrocytes (Supplementary Fig. 7). The data confirm the presence of S1P3+ astrocytes in the neuroinflammatory response of human brain metastases.

**In vitro analysis of S1P3 contribution to BBB/BTB function**. Overexpression of S1P3 in the most permeable brain metastases suggested the hypothesis that neuroinflammatory astrocytic S1P3 signaling may change BTB permeability. In vitro models of the BBB have been used to model in vivo function. Typically, brain endothelial cells were cultured on a porous membrane to confluency, and their transendothelial electrical resistance (TEER) or the passage of markers/drugs through the monolayer quantified as readouts of permeability[33,34]. To test the function of astrocytic S1P3 in the neuroinflammatory response, this assay was improved to incorporate multiple components of the BBB. Immortalized human brain endothelial cells were cultured to confluence on the luminal side of a filter, with immortalized human brain pericytes on the abluminal side. Immortalized human astrocytes were added at a distance at the bottom of the culture, similar to activated astrocytes surrounding a metastasis (Fig. 5a and Supplementary Fig. 8a–c). Assay validation experiments demonstrated the optimal conditions for attachment of the cells to the filter, the optimal arrangements of pericytes and astrocytes and correct localization of ZO-1 in the endothelial

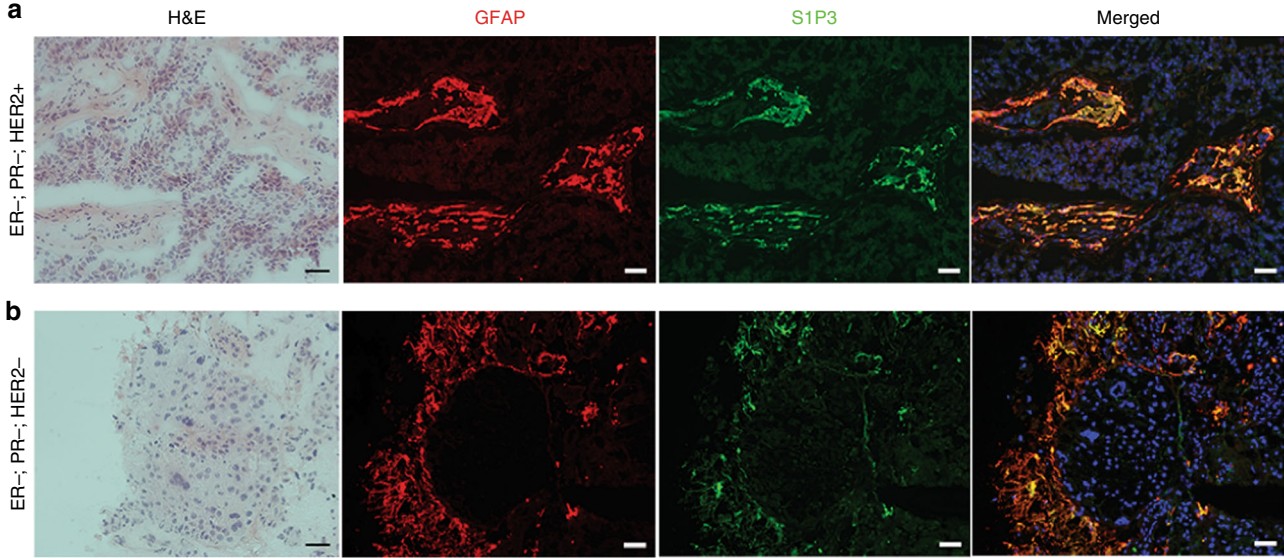

**Fig. 4** S1P3 is expressed in the neuroinflammatory response of patients' brain metastasis specimens. Fresh frozen sections of nine resected brain metastases of breast and lung cancers, nine of which were previously described[19], were stained for S1P3 (green) and GFAP (red). Nuclei are DAPI stained (blue). **a** Specimen #4, ER-, PR-, HER2+ breast cancer. **b** Specimen #5, ER-, PR-, HER2+ breast cancer. Co-staining of S1P3 and activated astrocytes in the tumor microenvironment is seen. A summary of all staining is presented in Supplementary Data 3. (Scale bar = 50 μm)

**Fig. 3** S1P3 overexpression in the astrocytic microenvironment of highly permeable brain metastases of breast cancer. **a** S1P3 is expressed by a proportion of activated astrocytes. Brain sections containing 231-BR experimental brain metastases were stained for S1P3 (green), GFAP+-activated astrocytes in the neuroinflammatory response (red), CD31 endothelial cells to visualize the location of astrocyte end feet (gray). All nuclei are stained with DAPI (blue). Right panel, merge. Scale bar: 50 μm. **b–e** Representative TRD permeability, S1P3, and GFAP staining in brain metastasis sections from four model systems (left); ratio of normalized astrocytic S1P3 in highly permeable/poorly permeable metastases (right). For each model system, distinctly permeable and poorly permeable metastases to TRD (red) were identified (scale bar = 100 μm); successive sections were stained for S1P3 (green) and GFAP (red) (scale bar = 50 μm). For quantification, S1P3 staining area was normalized to GFAP staining area. Staining in all distinctly highly permeable metastases/poorly permeable metastases in a single mouse brain was calculated (dot). The red line at unity would indicate no trend in highly/poorly permeable metastases. **b** Independent 231-BR cohort. Astrocytic S1P3 in highly/poorly permeable metastases, median = 1.73 ($n = 12$). **c** HER2+ JIMT-1-BR brain metastases[4]. Astrocytic S1P3 in highly/poorly permeable metastases, median = 1.36 ($n = 10$). **d** HER2+ inflammatory breast cancer SUM190-BR metastases[19]. Astrocytic S1P3 in highly/poorly permeable metastases, median = 1.80 ($n = 8$). **e** 4T1-BR brain metastases. Astrocytic S1P3 in highly/poorly permeable metastases, median = 1.59 ($n = 8$). From (**b**) to (**e**): Graphs represent the median, the interquartile range (box), and the min/max values (error bars). Wilcoxon matched-pairs signed rank test, two-tailed, *$P < 0.05$

layer (Supplementary Fig. 8d–f). To model a BTB, spheroids of 231-BR cells were added to the bottom of the cultures with the astrocytes (Supplementary Fig. 8g). In addition to electrical resistance, transport of doxorubicin through the BBB or BTB was quantified.

To assess the contribution and the specificity of S1P3 signaling, a panel of S1P receptor antagonists was evaluated in vitro (Fig. 5b). Validation experiments established an optimal dose without anti-proliferative effects (Supplementary Fig. 8h–i). An

S1P3 antagonist would be hypothesized to elevate the integrity of the BBB/BTB, and would serve as a proof of principle rather than a translational lead. Two antagonists targeting S1P3 significantly elevated TEER (Fig. 5c, d) and reduced doxorubicin permeability (Fig. 5e, f) in both BBB and BTB (tumor sphere containing) assays, whereas antagonists to S1P1, 2, and 4 were without effect. In the BBB model, TY-52156 increased TEER by 1.61 fold ($P = 0.0079$, Kruskal–Wallis test and Dunn's multiple comparison) while CAY10444 increased TEER by 2.28 fold ($P = 0.0079$). For

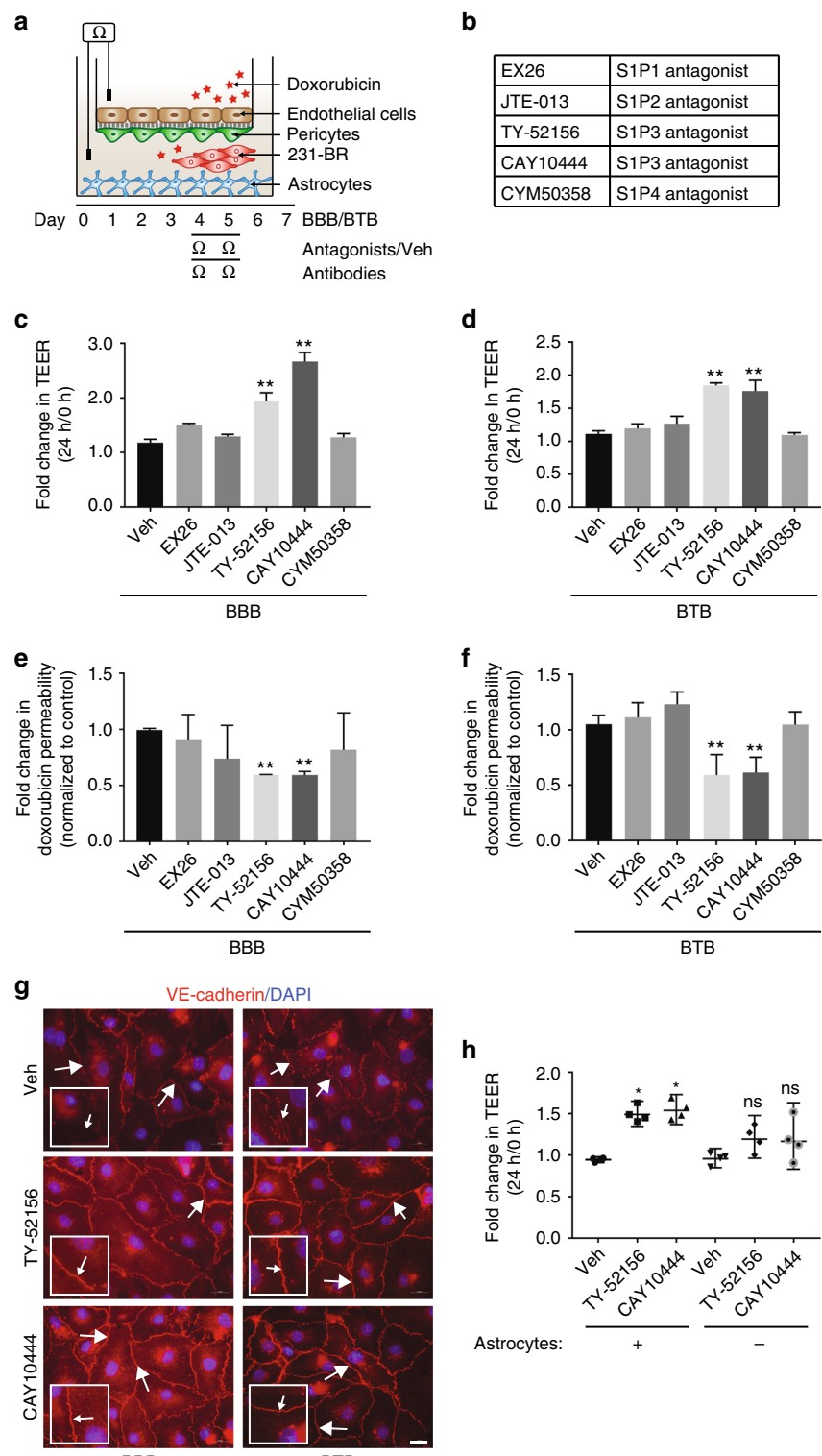

the BTB, TY-52156 increased TEER by 1.64 fold ($P = 0.0079$) and CAY10444 by 1.6 fold ($P = 0.0079$). In a second set of experiments, the permeability of the breast cancer chemotherapeutic doxorubicin decreased by 1.66 fold with TY-52156 ($P = 0.008$), 1.69 fold with CAY10444 ($P = 0.008$) in BBB, and 1.78 fold with TY-52156 ($P = 0.006$) and 1.72 fold with CAY10444 ($P = 0.006$) in BTB. Endothelial cells on the luminal side of the filter from the cultures were stained using IF at the experimental endpoint and showed increased expression and membranous localization of the cell–cell adhesion molecule VE-cadherin in response to TY-52156 and CAY10444 (Fig. 5g); similar trends were observed for the endothelial tight junction protein ZO-1 (Supplementary Fig. 8l). That TY-52156 and CAY10444 were working via astrocytes was demonstrated using cultures with and without astrocytes in the lower chamber. TY-52156 increased TEER by 1.55 fold and CAY10444 by 1.6 fold when astrocytes were present ($P = 0.028$, Kruskal–Wallis test and Dunn's multiple comparison). However, TEER increase by TY-52156 and CAY10444 was without significant effect in the absence of astrocytes (Fig. 5h).

A confirmatory approach used two independent sets of control shRNA and *S1PR3* shRNA transfected immortalized human astrocytes, the latter of which expressed less *S1PR3* mRNA and protein than shControl transfectants (Fig. 6a). When added to the lower chamber of BBB or BTB cultures, *S1PR3* shRNA transfectants increased the TEER by 1.33 fold in BBB ($P = 0.002$, Mann–Whitney test) and 1.43 fold in BTB ($P = 0.002$) for sh*S1PR3*-1, and 1.43 fold in BBB ($P = 0.002$) and 1.44 fold in BTB ($P = 0.002$) for sh*S1PR3*-2 (Fig. 6b). Fold change in doxorubicin permeability decreased by 1.68 fold and 1.59 fold in the BBB ($P = 0.0047$) and BTB ($P = 0.0002$) cultures respectively for sh*S1PR3*-1, and 1.71 fold and 1.66 fold in the BBB ($P = 0.0022$) and BTB ($P = 0.0022$) cultures respectively for sh*S1PR3*-2 (Fig. 6c). Knockdown of S1P3 (using sh*S1PR3*-1 and sh*S1PR3*-2) resulted in increased endothelial expression of VE-cadherin (Fig. 6d) and ZO-1 (Supplementary Fig. 8m) in both BBB and BTB cultures. Taken together, alterations in astrocytic S1P3 functionally modulated permeability in an in vitro model of the BBB and BTB.

**Pharmacological modulation of S1P3 in vivo.** In order to determine whether modulation of S1P3 altered the BBB/BTB in vivo, mice harboring 231-BR or SUM190-BR brain metastases were administered a limited course of brain-permeable TY-52156 by oral gavage, twice a day (b.i.d.) (Fig. 7a). Mice treated with $10 \text{ mg kg}^{-1}$ TY-52156 had detectable levels of compound in both plasma (range 550–650 ng per mL of plasma) and metastasis-containing brain (620–1135 pg per mg of brain) (Supplementary Fig. 9a). TY-52156 had no effect on the number of large (>300 μm

in a single direction) or micro-(smaller) metastases that developed in the 231-BR model (Supplementary Fig. 9b) nor on the number of metastases formed in the SUM190-BR model (Supplementary Fig. 9c, non-significant differences). In addition, proliferation was measured in the 231-BR model, using Ki67 normalized to human cytokeratin as a readout: TY-52156 showed no effect (Supplementary Fig. 9d). To measure the permeability, TRD was injected at the experimental endpoint and the animals perfused 10 min later. TRD intensity was normalized to human cytokeratin IF (as a measure of metastasis area) in metastases in one brain section per mouse. In vehicle-treated 231-BR metastasis-containing mice, TRD/cytokeratin area was quantified for 43 metastatic clusters and demonstrated typical heterogeneity, with a median of 0.171 (interquartile range (IQR) = 0.032–0.405). With administration of $10 \text{ mg kg}^{-1}$ TY-52156, heterogeneous permeability of metastases was evident, but at overall lower levels of TRD uptake. The TRD/cytokeratin ratio had a median of 0.052 (IQR = 0.006–0.18; $P = 0.016$, Two-tailed Mann–Whitney statistical test), showing a 70% decrease (Fig. 7b). As a second approach to quantification, TRD diffusion was also normalized to blood vessel density using CD31 staining. With vehicle treatment, the TRD/CD31 median was 0.353 (IQR = 0.109–0.778). Administration of TY-52156 decreased the TRD/CD31 ratio to a median of 0.121 (IQR = 0.007–0.277; $P = 0.012$) (Fig. 7c), showing a 66% decrease. For mice containing SUM190-BR metastases, TY-52156 treatment decreased the TRD/cytokeratin ratio by 74%, from 0.0498 (IQR = 0.023–0.184) to 0.0131 (IQR = 0.003–0.040; $P < 0.0001$) (Fig. 7d). TY-52156 treatment also decreased TRD diffusion when normalized to blood vessel density. TRD/CD31 ratio median decreased by 73%, from 1.333 (IQR = 0.682–5.972) to 0.3644 (IQR = 0.131–1.38; $P < 0.0001$) after TY-52156 treatment (Fig. 7e).

Subsequently, we asked if there was evidence of target modulation by TY-52156 in vivo. Even though this is a chemical inhibitor, there could be downstream effects on target expression. Interestingly, $10 \text{ mg kg}^{-1}$ TY-52156 decreased the S1P3 expression of GFAP+ astrocytes in the 231-BR model, suggesting a feedback loop where receptor inhibition influences protein expression level (vehicle median = 0.101, IQR = 0.054–0.18; $10 \text{ mg kg}^{-1}$ TY-52156 median = 0.059, IQR = 0.007–0.08; $P < 0.001$, Mann–Whitney test) (Supplementary Fig. 9e). Total activated astrocytes (GFAP) did not vary with treatment (Supplementary Fig. 9f). The data indicate that S1P3 inhibition functionally decreased the permeability of the BTB of experimental brain metastases.

To determine the effect of TY-52156 on drug uptake in an experimental metastasis assay, mice injected with the 231-BR cancer cells were randomized on day 25 post-injection to receive vehicle or TY-52156 twice daily (b.i.d.) for 4 days; on days 26 and

**Fig. 5** S1P3 antagonists strengthen the blood–tumor barrier in vitro. **a** An in vitro model of the BBB was constructed with confluent brain endothelial cells on the luminal side of a coated porous membrane, brain pericytes on the abluminal side of the membrane, and astrocytes in the bottom of the abluminal side of the chamber. To create a blood–tumor barrier (BTB) model, spheres of 231-BR tumor cells were added to the bottom of the chambers. Barrier integrity was quantified by transendothelial electrical resistance (TEER), or diffusion of a drug (doxorubicin) from the luminal to abluminal side. Cells were cultured to confluence for 4 days, at which time antagonists (**b**) (EX26: 1 nM, JTE-013: 17 μM, TY: 2 μM, CAY10444: 10 μM, and CYM50358: 25 nM) or vehicle were added. In BBBs and BTBs, both S1P3 antagonists selectively showed increased TEER, indicative of a tighter BBB (**c**) and BTB (**d**) ($n = 5$). TY-52156 and CAY10444 decreased permeability of 100 μg mL$^{-1}$ doxorubicin significantly in BBB (**e**) and BTB (**f**) models ($n = 5$). **g** After 24 h post-treatment with TY-52156 and CAY10444, the membranes containing brain endothelial cells were removed and stained for VE-cadherin (red) (white arrows). Nuclei were DAPI stained (blue). Increased staining and membranous localization was observed with TY-52156 and CAY10444 (scale bar = 20 μm). **h** The effect of vehicle, 2 μM TY-52156 or 10 μM CAY10444 on BBB TEER in cultures with or without astrocytes. Significant TEER elevation by TY-52156 and CAY10444 was dependent on the presence of astrocytes ($n = 4$). For (**c**-**f**) and (**h**), Kruskal–Wallis test and Dunn's multiple comparison test were performed. Graphs represent the median and the error bars the 95% confidence interval, from independent transwell BBB/BTB cultures ($n$) from multiple experiments. *$P < 0.05$, **$P < 0.01$, ns non-significant

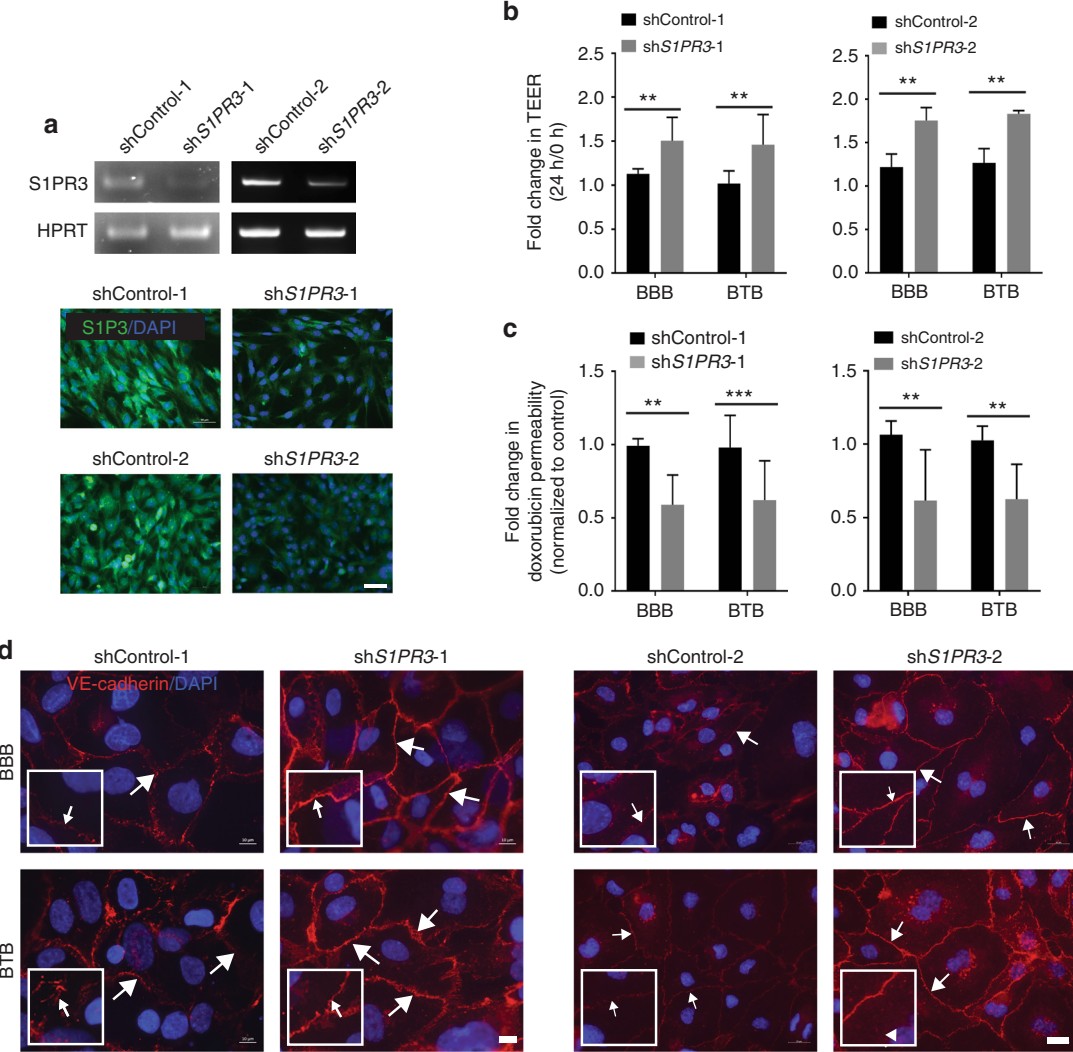

**Fig. 6** Knockdown of astrocytic S1PR3 strengthened the BBB and BTB in vitro. Immortalized human astrocytes were transduced with either control shRNA (shControl-1, shControl-2) or shS1PR3 (shS1PR3-1, shS1PR3-2), and used in TEER and permeability assays of the BBB and BTB (see legend to Fig. 5). **a** RT-PCR for S1PR3 mRNA (top) and immunofluorescent staining (bottom) for S1P3 protein (green). Nuclei are DAPI stained (blue) (scale bar = 50 μm). **b** TEER assays using either shControl-1, -2 or shS1PR3-1, -2 astrocytes in the cultures. S1PR3 knockdown elevated TEER at 24 h, indicative of a tighter BBB and BTB, (n = 6). **c** Permeability of 100 μg mL$^{-1}$ doxorubicin from luminal to abluminal side of the inserts in the BBB and BTB, (n = 6). Astrocytic S1P3 knockdown (using shS1PR3-1 and shS1PR3-2) reduced permeability. **d** Staining of endothelial cell monolayers in TEER assays for VE-cadherin (red) demonstrating increased membrane staining (white arrows) when shS1PR3-1, -2 astrocytes were included. Nuclei were DAPI stained (blue) (scale bars = 10 μm for set-1 and 20 μm for set-2). For (**b**) and (**c**), Mann–Whitney statistical analysis was performed. Graphs represent the median and the error bars the 95% confidence intervals, from independent transwell BBB/BTB cultures (n) from multiple experiments. **P < 0.01, ***P < 0.001

28 post-injection, all mice received a dose of 50 mg kg$^{-1}$ methotrexate (MTX). TRD was administered before necropsy followed by perfusion. MTX was selected as it readily crosses the BBB and is not a substrate of efflux pumps that may confound interpretations. Endpoints were MTX levels in whole brain homogenate/plasma. The brain measurements differ from TRD experiments above in that they include normal brain and all lesions. A 30% reduction in [MTX]$_{brain/plasma}$ was observed in response to TY-52156, a statistical trend (Supplementary Fig. 10).

**Modulation of BTB permeability by S1P3-induced cytokines.** TY-52156 demonstrated the functionality of astrocytic S1P3 in modulation of the BTB, but works in a direction opposite of a therapeutic goal in oncology. Agonists of S1P3 have shown cardiac toxicity[35]. We investigated the mechanism of action of astrocytic S1P3 in the hope that a translational lead would emerge

from its functional pathway. Based on the observation that astrocytes in the bottom of the TEER chambers altered multiple facets of the distant brain endothelial cells, we hypothesized that modulation of astrocytic S1P3 expression alters its secretory phenotype. Culture supernatants of shS1PR3-1-transfected immortalized astrocytes showed reduced levels of multiple chemokines, growth factors, and interleukins as compared to shControl-1 astrocytes (Fig. 8a). Similar results were observed when independent set of S1P3 knockdown supernatants from shControl-2 and shS1PR3-2 were analyzed (Supplementary Fig. 11a). To test whether differences in the levels of one or more astrocyte-secreted proteins was contributory to regulation of the BBB, neutralizing antibodies, or a control isotype-matched unrelated antibody were added to the astrocytes in the lower chambers of BBB cultures. Only antibodies to IL-6 and CCL2 significantly altered culture permeability using both TEER and doxorubicin permeability as readouts, with no significant

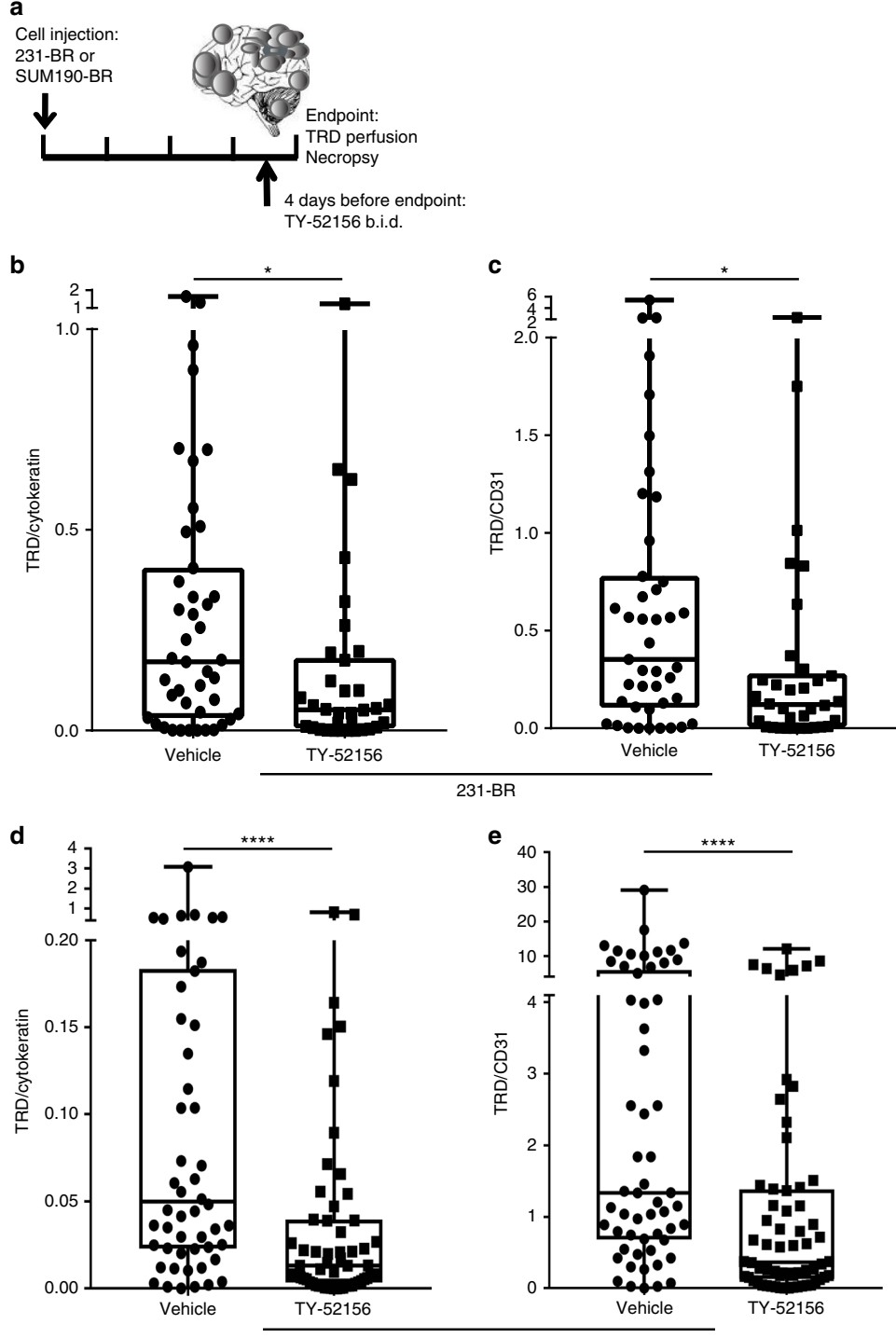

**Fig. 7** TY-52156 strengthens the blood–tumor barrier in vivo. **a** Mice harboring experimental brain metastases from the 231-BR or the SUM190-BR received 10 mg kg$^{-1}$ TY-52156 or vehicle, b.i.d. by oral gavage, for 4 days. On the fourth day of treatment, mice were injected with TRD and perfused 10 min later. Brain and blood were taken at necropsy. **b–e** TRD pictures were taken, subsequently the same tissue slides were stained for human cytokeratin or CD31 to normalize TRD diffusion to either metastasis cellularity or blood vessel density, respectively. **b,c** Graphs represent metastatic clusters from 11 and eight mice for vehicle and 10 mg kg$^{-1}$ TY-52156, respectively. **b** TRD/cytokeratin intensity in 231-BR brain metastatic clusters. **c** TRD/CD31 intensity in 231-BR brain metastatic clusters. **d,e** Graphs represent metastatic clusters from five and eight mice for vehicle and 10 mg kg$^{-1}$ TY-52156, respectively. **d** TRD/cytokeratin intensity in SUM190-BR brain metastatic lesions. **e** TRD/CD31 intensity in SUM190-BR brain metastatic lesions. **b–e** One dot represents one cluster of metastases in the 231-BR model and one lesion in the SUM190-BR model. Graphs represent the median, the interquartile range (box), and the min/max values (error bars). Two-tailed Mann–Whitney statistical analysis; *$P < 0.05$; ****$P < 0.0001$

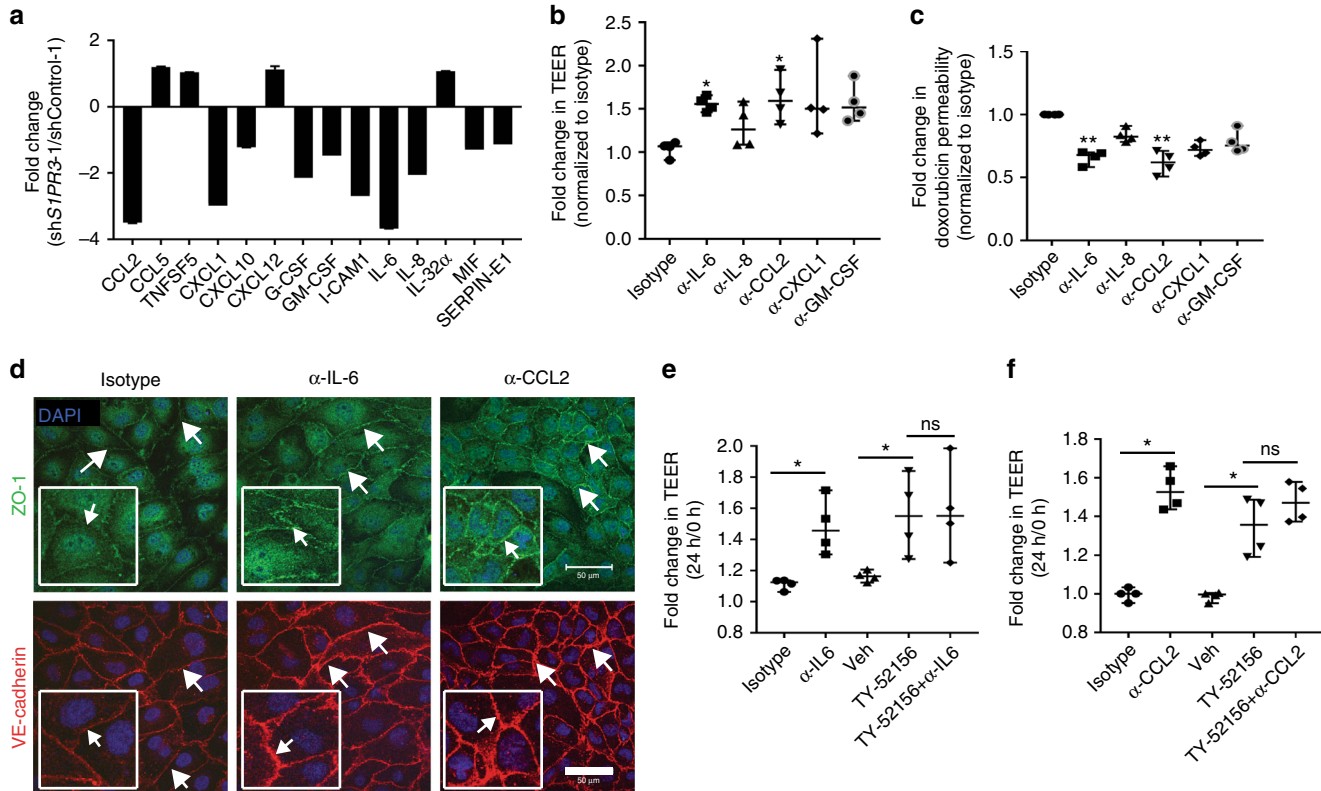

**Fig. 8** Knockdown of S1PR3 in astrocytes reduced cytokine secretion and altered endothelial adhesion. Serum-free culture supernatant from shControl-1 and sh*S1PR3*-1 astrocytes were compared for expression of 36 cytokines and chemokines using a commercial human cytokine array. **a** Mean spot pixel density was measured of quantifiable spots with ImageJ and plotted which demonstrated a significant reduction in various cytokines (two spots for each cytokine). **b** The effect of neutralizing antibodies to cytokines on BBB permeability in vitro. See Fig. 5a for experimental schematic. Addition of neutralizing antibodies to IL-6, IL-8, CCL2, CXCL1, and GM-CSF increased TEER resistance as compared to an isotype-matched control antibody. Only anti-IL-6 and anti-CCL2 at 24 h showed a significant increase in TEER ($n = 4$). **c** Neutralizing antibodies to cytokines decreased doxorubicin permeability. Permeability of 100 µg mL$^{-1}$ doxorubicin from luminal to abluminal side of the inserts in the BBB was measured. Anti-IL-6 and anti-CCL2 decreased doxorubicin permeability significantly ($n = 4$). **d** Staining of endothelial cells for ZO-1 (green) and VE-cadherin (red) in BBB cultures treated with neutralizing antibodies. Nuclei were DAPI stained (blue) (scale bar = 50 µm). The effect of combined treatment of astrocytes with TY-52156 and anti-IL-6 antibody or anti-CCL2 antibody on TEER. There was no significant augmentation of resistance when IL-6 antibody (**e**) or CCL2 (**f**) antibody were included along with TY-52156 in BBB model ($n = 4$). For panels (**b**), (**c**), (**e**), and (**f**), each point indicates median fold change, and the bars represent the overall median with the 95% confidence interval, for each BBB culture. Kruskal–Wallis test and Dunn's multiple comparison test were performed. *$P < 0.05$, **$P < 0.01$, ns non-significant

contributions of antibodies to IL-8, CXCL1, and GM-CSF (Fig. 8b, c). Neutralizing antibodies to IL-6 and CCL2 intensified endothelial ZO-1 and VE-cadherin expression (Fig. 8d) in agreement with TY-52156 and sh*S1PR3* data. For CCL2, multiple shRNAs to S1P3 downregulated its secretion by astrocytes (Supplementary Fig. 11b). S1P3-specific inhibitors downregulated astrocytic CCL2 secretion (TY-52156 by 1.37 fold, $P = 0.0286$, CAY10444 by 1.57 fold, $P = 0.0286$, Kruskal–Wallis test and Dunn's multiple comparison test), while inhibitors to other S1P receptor family members were without effect (Supplementary Fig. 11c). No additivity or synergy was observed between neutralizing antibodies and TY-52156 in altering BBB permeability, consistent with both lying on the same functional pathway (Fig. 8e, f).

**Tumor cell overexpression of S1P3.** Given the cardiotoxicity of a S1P3 agonist, we asked if tumor cells could serve as "Trojan horses" to bring S1P3 signaling behind the BTB. Brain-tropic 231-BR cells expressing S1P3 recapitulated the chemokine profile of neuroinflammatory astrocytes, expressing greater levels of CCL2 and IL-6 than Vector transfectants (Fig. 9a, b). When injected into mice, S1P3 overexpressing tumor cells produced 59% greater micrometastases (231-BR-Vector median = 350,

IQR = 258–458; 231-BR-S1P3 median = 557, IQR = 378–611; $P = 0.0374$, Two-tailed Mann–Whitney statistical test) and 44% greater large metastases (the latter was not significant: 231-BR-Vector median = 6.9, IQR = 4.15–8.3; 231-BR-S1P3 median = 12.33, IQR = 5.2–13.4; $P = $ ns) (Fig. 9c, d). Mice were administered TRD and perfused before necropsy to visualize metastasis permeability, which was normalized to human tumor cytokeratin content by IF staining. Both Vector and S1P3 overexpressing lines produced metastases with heterogeneous permeabilities (Fig. 9e), however fewer metastases of very low permeability were evident in the S1P3 overexpressing 231-BR cells. Using quartiles of TRD/cytokeratin permeability, the percentage of low permeability metastatic clusters was 36% in the vector transfectants and 15% in the S1P3 overexpressing tumor cells ($P = 0.0382$, Chi-square test) (Fig. 9f). Conversely, the percentage of high permeability metastatic clusters was 64% in the vector transfectants and increased to 85% in the S1P3 overexpressing tumor cells. The data provide proof of principle that S1P3 signaling can functionally modulate BTB permeability.

**Discussion**
At least three avenues of research are attempting to improve drug efficacy for brain metastases: (1) identification of new

actionable targets[36–39], (2) development of brain-permeable drugs and treatments[40–42], and (3) understanding the structure and function of the BTB to identify permeability-modulating pathways[19,43,44]. With reference to the BTB, it has been largely unknown if this structure is just a random breakdown of the BBB or, alternatively, has consistent features. If so, the BTB could serve as a tractable target to improve therapeutic permeability. Herein we focus on those experimental brain metastases that are most permeable to identify potential pathways for translational development.

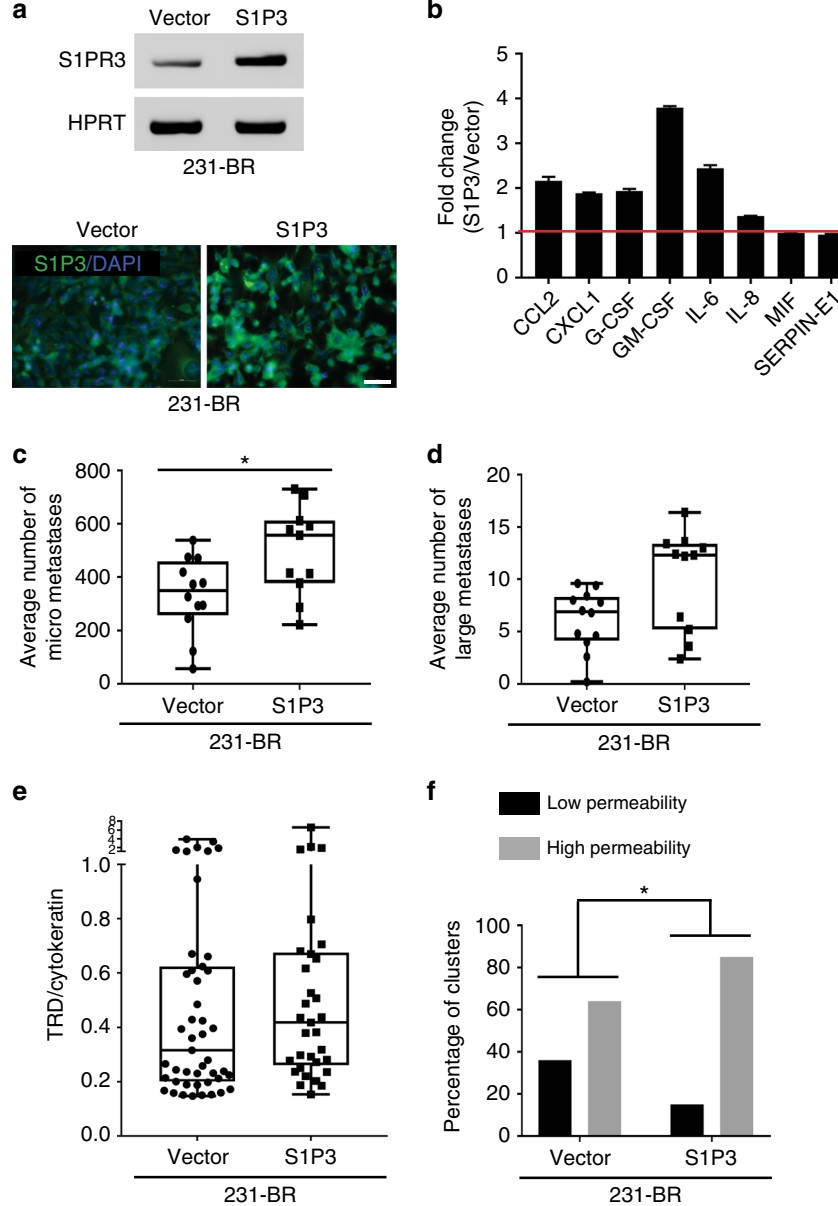

**Fig. 9** Overexpression of S1P3 in tumor cells. Brain-tropic 231-BR cells were stably transduced with vector alone or S1P3 lentiviruses. **a** RT-PCR for *S1PR3* mRNA (top) and immunofluorescent staining (bottom) for S1P3 protein (green). Nuclei are DAPI stained (blue) (scale bar = 50 μm). **b** Serum-free culture supernatant from Vector and S1P3 overexpressing 231-BR cells were compared for expression of 36 cytokines and chemokines using a commercial human cytokine array. Mean spot pixel density was measured of quantifiable spots with ImageJ and plotted, which demonstrated an increase in various cytokines (two spots for each cytokine, red line indicates no change). **c–f** Mice were randomized to receive either 231-BR-Vector (*n* = 12) or 231-BR-S1P3 (*n* = 11) overexpressing cells. The metastases grew for 4 weeks. At the end point, mice were injected with TRD and perfused 10 min later. Brains were taken at necropsy. **c,d** Each dot represents the number of micro- or large metastases per mouse. The 231-BR-S1P3 line significantly increased the number of micrometastases (**c**) and shows a trend in increasing the number of large metastases (**d**). **e** TRD diffusion was analyzed by normalization to metastasis cellularity using human cytokeratin staining. A 32% increase in the ratio TRD/cytokeratin median was observed in the 231-BR-S1P3 injected mice (*n* = 8) compared to the 231-BR-Vector injected mice (*n* = 10) but it did not reach statistical significance. Each dot represents a metastatic cluster. Graphs represent the median, the interquartile range (box) and the min/max values (error bars) (231-BR-Vector median = 0.316, IQR = 0.201–0.624; 231-BR-S1P3 median = 0.418, IQR = 0.2612–0.675; *P* = ns). Two-tailed Mann–Whitney statistical analysis (**c–e**). **f** The distribution of the TRD/cytokeratin measurement for all clusters was analyzed: the 25% percentile (TRD/cytokeratin = 0.226) was used as a cut off to dichotomize the distribution in low vs. high permeability. Graphs represent the number of clusters in each category (**f**). Chi-square statistical analysis was performed to analyze the interaction between S1P3 overexpression and cluster permeability (low vs. high). *$P < 0.05$

The gene expression profile of highly permeable experimental metastases was compared to poorly permeable lesions in the same brain. The salient technical features of our approach included mouse model systems of hematogenous brain metastasis of breast cancer. A 3 kDa TRD served as a measure of paracellular permeability, as its uptake into brain metastases directly correlated with that of paclitaxel, doxorubicin, lapatinib, and vinorelbine[4,10,20]. A new use for LCM was developed to enrich brain metastases, isolating independently highly vs. poorly permeable lesions, including both human tumor cells and the mouse brain microenvironment. Consistent changes in gene expression were identified, mostly in the mouse (microenvironmental) arrays. Validation studies at the protein level identified activated astrocytes in the neuroinflammatory response as the major source of the differential expression. Astrocytes form a network throughout the normal brain, regulating ion homeostasis, neurotransmitter clearance, and synapse and neurovascular coupling formation[45]. Astrocytes make two contributions to brain metastases: their endfeet contribute to the BBB, and activated astrocytes physically surround metastases together with activated microglia and immune cells to form the neuroinflammatory response. The neuroinflammatory response is a part of multiple CNS diseases, with both disease-limiting and promoting roles[46]. In highly permeable 231-BR metastases neuroinflammatory astrocytes were S1P3[high], EphA5[low], GABA A receptor[low]. These data add to the growing literature concluding that a consistent set of alterations are found in the BTB, rather than random breakdown of the BBB. Importantly, in 17/19 frozen human craniotomy specimens, S1P3 expression was detected in GFAP+ reactive astrocytes. Reactive astrocytes from the most permeable metastases were also morphologically thin and truncated as compared to more traditional hypertrophic astrocytes with large soma and multiple star-shaped processes in normal brain and low permeability metastases. The astrocytes accompanying highly permeable lesions appeared similar to the activated radial scar border astrocytes described in multiple CNS insults and Alzheimer's disease[46,47]. These data complement a growing literature of other astrocytic contributions to brain metastases[48–51]. It remains possible that other differentially expressed genes were masked by an overall low, or varying contributions of a cell type to the entirety of the brain metastasis.

S1P3, one of a family of receptors for the lipid mediator S1P, was overexpressed by 40–89% in neuroinflammatory astrocytes surrounding the most permeable metastases (as compared to the least permeable metastases) in four model systems of brain metastasis of breast cancer. Astrocytic S1P3 expression has been reported in other brain diseases involving BBB permeability alterations, MS[30,31], and Sandhoff disease[52], suggesting that this pathway may be more generalized. Both S1P3-specific antagonists and shRNA *S1PR3* knockdown in immortalized astrocytes tightened the BBB and BTB in an in vitro model with TEER and drug permeability as endpoints; antagonists to other S1P receptor family members were not functional. Tightening of the BBB and BTB was associated with phenotypic changes in the brain endothelial cells, specifically, in the amount and localization of tight junction and adhesion proteins. The in vitro BBB model system offers the opportunity to interrogate specific sets of cells under multiple experimental conditions; despite years of optimization it remains an incomplete correlate of the actual BBB. In vivo, a limited course of S1P3 antagonist treatment also significantly tightened the BTB in two model systems, but was without significant effects on the already closed BBB. Permeability of methotrexate was also quantified in whole brain homogenates after S1P3 antagonist treatment, and demonstrated a trend of increased permeability. These data are encouraging as the whole brain homogenates were dominated by uninvolved brain, where

the TRD experiments only quantified the metastatic lesions. Our findings provide a proof of principle that compounds targeting the astrocytic neuroinflammatory response in the brain metastatic microenvironment can modulate BTB permeability. While these data served as a proof of principle for brain metastases, the therapeutic potential of TY-52156 in CNS diseases characterized by BBB dysfunction, such as MS, traumatic brain injury, stroke, etc. may be of interest.

An in vitro downstream pathway was uncovered, whereby S1P3 signaling affects BTB permeability. Based on the observation that astrocytes distant from the endothelial-pericyte cultures altered permeability in an S1P3-dependent manner in vitro, we hypothesized that secreted products may be involved. A sampling of astrocyte-secreted chemokines, growth factors, and interleukins revealed several factors downregulated by S1PR3 knockdown. Neutralizing antibodies to CCL2 and IL-6 consistently tightened the in vitro BTB to a comparable extent as S1P3 inhibition in vitro. They did not appear to be additive or synergistic, suggesting a common pathway. Chemokine, cytokine, or interleukin secretion by activated astrocytes is an early event in non-cancer CNS diseases with altered BBB permeability, such as traumatic injury and autoimmunity (rev. in[53]). Both CCL2 and IL-6 have been functionally implicated in BBB permeability in non-cancerous CNS diseases[54–56], typically accompanied by endothelial tight junction changes.

Translation of these findings could utilize multiple approaches. To date, a S1P3 agonist is untranslatable due to cardiotoxic effects[35]. Altering the neuroinflammatory response may stand as one potential point of intervention. For astrocyte-derived chemokines and growth factors, the goal would be to elevate their expression on the abluminal side of the BTB rather than in systemic circulating levels. Each of these cytokines would be expected to produce complex phenotypes in addition to BBB/BTB permeability, including myeloid, dendritic cell and lymphocyte recruitment[55,57–59], neuronal and neural stem cell function[60–62], and cognition[63]. A beneficial drug permeability effect of CCL2 or IL-6 must outweigh potential tumor stimulatory pathways[64,65]. Lung cancer lines including a brain-derived variant were insensitive to IL-6, but the extent of this resistance profile is unknown[66].

Experiments in which 231-BR tumor cells were engineered to overexpress S1P3 provide a proof of principle for this potential therapeutic pathway. Here, the 231-BR S1P3 overexpressing cells extravasated to the abluminal side of the BTB and remarkably recapitulated the cytokine signaling profile of the neuroinflammatory astrocytes. Two phenotypes emerged: one concerning the micrometastasis growth and the other concerning the TRD diffusion pattern. Increased micrometastatic lesions were produced by the S1P3 overexpressing cells. The range of TRD permeability did not change as compared to controls, but median permeability of S1P3 overexpressing tumor cells was increased by 31%. Within the range of permeabilities, significantly greater number of 231-BR S1P3 overexpressing lesions were elevated out of the lowest quartile of permeability. Future experiments will determine if different levels of S1P3 overexpression or time courses will more profoundly alter permeability, and if increased permeability of specific drugs outweighs increased micrometastasis formation.

Another point of potential translational control is the brain endothelial cells. Using both two S1P3 antagonists and sh*S1PR3* constructs, modulation of permeability was accompanied by altered brain endothelial levels and localization of ZO-1 in tight junctions and the VE-cadherin cell adhesion factor in vitro. This potential point of intervention offers the benefit of accessibility. While the transcriptome and secretome of BBB endothelial cells has been reported[67,68], additional investigation of BTB

endothelia are needed to fully understand the list of tractable S1P3 targets.

## Methods

**Cell line origin and authentication.** The MDA-MB-231 BR "brain-seeking" variant (231-BR) was provided by Dr. Toshiyuki Yoneda, Indiana University School of Medicine, Indianapolis, IN, USA. The 231-BR line was subsequently transfected with eGFP and a quantifiable brain metastasis assay was developed[17]. The parental JIMT-1 line was provided by Dr. Dennis J. Slamon, UCLA Jonsson Comprehensive Cancer Center, Los Angeles, CA, USA. The brain seeking variant of the JIMT-1, the JIMT-1-BR, was developed after three rounds of intracardiac injection, brain dissection, cell culture, and re-injection in the left cardiac ventricle of the mice[18]. The SUM190 breast cancer cell line was obtained from Asterland (Detroit, MI). Similar to the JIMT-1-BR, the brain derivative, SUM190-BR, was developed after three rounds of intracardiac injection/brain cell culture[19]. The 4T1-BR cell line was provided by Dr. Suyun Huang, MD Anderson Cancer Center, TX, USA. Parental lines MDA-MB-231, SUM190, and JIMT-1 were sent to ATCC® for authentication, using 17 short tandem repeat (STR) loci and the gender determining locus. Of these, only MDA-MB-231 was available in the ATCC® STR database and was a match. The SUM190 and JIMT-1 lines were not in the ATCC® STR database and could not be authenticated. Each brain seeking variant was compared to the parental line using the same technology and all the three matched their corresponding parental line, confirming the provenance of the brain seeking variants.

**Animal experiments.** Animal experiments were conducted under a National Cancer Institute-approved Animal Use Agreement, following the Animal Care and Use Committee (ACUC) regulations. Female athymic NIH nu/nu mice were purchased from Charles River Laboratories. At 5–7 weeks old, mice were anesthetized with isoflurane/$O_2$ and tumor cells were injected in the left cardiac ventricle each in 0.1 mL $Ca^{2+}$ $Mg^{2+}$ free PBS: (a) MDA-MB-231-BR cells (231-BR) ($1.75 \times 10^5$ cells, total assay time approximately 4 weeks), (b) SUM190-BR cells ($5 \times 10^5$ cells, total assay time approximately 10 weeks) or (c) JIMT-1-BR cells ($1.75 \times 10^5$ cells, total assay time approximately 3 weeks). The 4T1-BR cells ($5 \times 10^4$ cells, total assay time approximately 2 weeks) were injected in immune-competent BALB/c mice (Charles River Laboratories). Before injection of mice, all cell lines were tested for a panel of viruses, per NIH ACUC regulations. The cells were negative for Mycoplasma and all the viruses tested. The left cardiac ventricle injection is performed manually by trained animal technicians. It is expected to have 5% misinjection. The mice that did not develop any metastases due to misinjection were excluded from the studies. The metastatic models are heterogenous and some mice will develop metastatic burden faster than others. In accordance to the NIH ACUC regulations, animals are euthanized when they appear moribund, cachectic, or unable to obtain food or water. To account for the misinjected mice and the mice that developed early morbid symptoms and required euthanasia, 10–12 mice were injected per group, to have approximately eight mice per group for final analysis. The procedure for TRD injection was previously described[19]. Briefly, at the end of the experiment, a tail vein injection of 1.5 mg mL$^{-1}$ 3 kDa TRD in 100 µl of 0.9% NaCl was delivered. While the marker circulated for 10 min, mice were settled under deep anesthesia, and were subsequently perfused with Krebs–Ringer Bicarbonate buffer for 1 min to washout TRD from the vasculature. Brains were dissected at necropsy, embedded in OCT and frozen in an ethanol/dry ice bath. Animal care (drug treatments and necropsy), brain sectioning/slide labeling, and IF staining/analysis were performed by three different individuals. The investigator performing all the analyses (metastasis count and all the IF staining) was blinded to the group assignment.

**Laser capture microdissection.** LCM was combined with fluorescence microscopy to identify and isolate highly vs. poorly permeable metastases. OCT-embedded frozen brains were sectioned at 8 µm in a cryostat and mounted on Arcturus® polyethylene naphthalate (PEN) membrane frame slides (metal framed) (ThermoFisher Scientific) to perform LCM. All sections were stored at −80 °C. Every other tissue section was directly observed for TRD exudation. The intervening frozen section was used to selectively microdissect and independently isolate highly permeable vs. poorly permeable metastases within a single brain, including their respective immediate brain microenvironments. The Arcturus XT™ LCM system and Arcturus® CapSure® Macro LCM caps were used to collect the LCM tissues. Approximately 5–10 sections per mouse brain were microdissected. Each of the highly permeable lesions, and each of the poorly permeable lesions were combined. After dissection, RNA was immediately extracted from the caps using the PicoPure® RNA isolation kit (ThermoFisher Scientific) following manufacturer's protocol. Extracted RNA was stored at −80 °C.

**RNA extraction and microarray analysis.** Quality and quantities of RNA were analyzed using the RNA Pico Chips and the Agilent 2100 Bioanalyzer (Agilent Technologies). Each sample had an average of 117 ng µL$^{-1}$ (10 µL total) and a RNA integrity number of 8.5. Gene expression profiling was performed by the NCI-Frederick Genomics Core Facility using the Nugen Ovation® labeling kit in conjunction with the following GeneChip®: Human Genome U133 Plus 2.0 array (whole human genome expression array for over 47,000 transcripts including ESTs) and Mouse Genome 430 2.0 array (whole mouse genome expression array for over 39,000 transcripts including ESTs) (Affymetrix). A pivot table was generated for the human and the mouse GeneChip, with t-test analysis used to compare data from highly permeable vs. poorly permeable lesions. Genes with P-values of <0.05 were prioritized based on the following four-parameter scoring system: (1) the magnitude of the differential expression (higher fold-changes were prioritized), (2) the fraction of total variance explained by "between group" differences (after partitioning the total expression variance for each gene into "within group" and "between group" differences, higher "between group" differences were prioritized), (3) the modulation of one group's expression beyond the median of the other (clear separation between groups, independent of fold-change magnitude, was prioritized), and (4) "within group" correlation of expression level and permeability (genes whose expression correlated with the degree of permeability within the permeable metastases group were prioritized) (Supplementary Data 1 and 2). Heat maps were generated using Matrix2png version 1.2.1 software.

**Validation of the LCM/microarray.** For each mouse of a new cohort, one 8 µm brain section was analyzed directly under a fluorescent microscope to identify highly vs. poorly TRD permeable metastases. The adjacent section was stained with antibodies directed to the proteins of interest: GFAP (1:10,000—cat#MAB360, Millipore), GABA A receptor α1, (1:50—cat#75–136 clone N95/35, NeuroMab), GABA A receptor γ2 (1:50—cat#224003, Synaptic System), S1P3 (1:100—cat#LS-B2155, LSBio and 1:100—cat#NBP1-00789, Novus), EphA5 (1:50—cat#GTX25398, GeneTex Inc.), NeuN (1:50—cat#MAB377, Millipore), and CD31 (1:500—cat#550274, BD Pharmingen). See Supplementary Table 1 for all antibodies' references. Briefly, slides were fixed with methanol for 5 min at −20 °C, washed with PBS then incubated in blocking buffer (PBS with 5% goat serum, DAKO) for 20 min at room temperature. Primary antibodies were incubated overnight at 4 °C. After three washes, the secondary antibodies (1:500, Alexa Fluor® antibodies) and DAPI were incubated for 1 h at room temperature. The slides were mounted using fluorescence mounting medium (Dako). Photographs of highly permeable and poorly permeable metastatic lesions in each section were acquired with the Zeiss Axioskop, at 10× or 20×. Quantification of immunostaining was performed on the entire field with the analysis program of Zeiss Axioskop Axiovision4 © software as previously described[19]. GABA A receptor α1, GABA A receptor γ2, and S1P3 antibodies stained activated astrocytes, therefore the quantification of these three markers was normalized to GFAP staining. EphA5 antibody stained astrocytes, neurons, and at a lesser extent metastatic cells. There was no marker that could represent those three cellular components, therefore the quantification of EphA5 staining was expressed as percentage of area stained in the entire microscopic field.

**Image analysis and figure processing.** All immunofluorescent pictures were acquired with the Zeiss Axioskop, and quantification performed on the original Zeiss format ".czi", using Zeiss Axioskop Axiovision4© software. For quantification, the surface area covered by each fluorescent marker was measured in the entire picture. The markers expressed in astrocytes (S1P3, GABA Aγ2, GABA Aα1, and EphA5) were normalized to all the activated astrocytes using GFAP. The maker expressed in neurons (EphA5) was normalized to all the neurons using NeuN. After quantification, all the ".czi" images were exported as jpeg/tif. All the photographs from the same experiment were adjusted for contrast and brightness in the exact same way in Photoshop®; e.g., the brightness and contrast parameters were the same for all the highly and poorly permeable metastatic clusters in one experiment. The different channel pictures were organized in a single canvas in Photoshop® before being copied into Illustrator®. All the figures were designed in Illustrator®.

**Synthesis of TY-52156.** This material was synthesized according to the method described by Murakami et al. with minor modifications [69]. Starting materials and solvents were purchased from commercial sources and used without further purification. Flash column chromatography was performed on a CombiFlash Rf 200i (Teledyne Isco Inc.) using silica cartridges from that manufacturer. $^1$H and $^{13}$C NMR data were collected on a Bruker 400 MHz spectrometer and are reported as ppm relative to the deuterated solvent signal. $^1$H NMR data are reported as chemical shift (δ ppm), multiplicity, coupling constant (Hz) and integration. $^{13}$C NMR data are reported as Chemical shift (δ ppm). Nominal mass LC/MS was performed on a Shimadzu LCMS-2020 single quadrupole instrument using a Kinetics 2.6 µ C18 100 Å (2.1 mm × 50 mm) column from Phenominex Inc. Solvent A = water, 0.1% formic acid, solvent B = acetonitrile, 0.1% formic acid. Four-minute gradients from 10% A to 90% B were employed for all products.

5,5-Dimethyl-2,4-dihexanone (1): Potassium tert-butoxide (54 g, 481 mmol) was suspended in diisopropyl ether (0.5 L) under argon and cooled in an ice bath. Pinacolone (25 g, 250 mmol) which had been dissolved in 75 mL of ethyl acetate was added dropwise with fast stirring. Following addition, the ice bath was removed and the reaction was allowed to warm to room temperature with stirring for 18 h. The reaction was returned to an ice bath and quenched slowly with 160 mL of water. The organic phase was removed and extracted with 25 mL of 1 M sodium hydroxide. The combined aqueous phases (quenching water and sodium hydroxide solution) were combined and acidified with 125 mL of 6 M HCl. The

acidified aqueous phase was extracted twice with 125 mL portions of hexanes. The combined hexanes were washed with 75 mL of water and then with 75 mL of brine. The hexanes were then dried over sodium sulfate and the solvent removed under vacuum at room temperature. The material was purified by vacuum distillation at approximately 10 mbar and 50 °C. This gave 23.72 g (167 mmol, 67%) of the desired product (1). $^1$H NMR (400 MHz, CDCl3) δ 5.61 (s,1H), 2.08 (s, 3H), 1.17 (s, 9H) reflecting the enolate; LC/MS $M/Z$ = 143.1 (M+H)+ (calculated), $M/Z$ = 143.3 (M+H)+ (found).

3-Chloro-5,5-dimethyl-2,4-dihexanone (2): 5,5-Dimethyl-2,4-dihexanone (23.7 g, 167 mmol) was dissolved in 450 mL of chloroform under argon and placed in an ice bath. Sulfuryl chloride (17.5 mL, 217 mmol) which had been diluted with 9 mL of chloroform was added dropwise with stirring. The ice bath was removed after addition and the reaction stirred 2 h. This is a deviation from the protocol described by Murakami et al., which claims the same stoichiometry as described here but reports to have used 10-fold more sulfuryl chloride in chloroform solution. The reaction was returned to the ice bath and quenched with 350 mL of water. The aqueous phase was discarded and the organics were washed three times with 350 mL portions of water. The organics were dried over sodium sulfate and the solvent was removed under vacuum at room temperature. The material was purified by vacuum distillation at approximately 2 mbar and 45 °C. This gave 26.1 g (148 mmol, 89%) of the intended product (2). $^1$H NMR (400 MHz, CDCl3) δ 5.10 (s, 1H), 2.38 (s,3H), 1.24 (s, 9H); LC/MS $M/Z$ = 175.1 (M−H)− (calculated), $M/Z$ = 175.1 (found).

[1-Chloro-1-(4-chlorophenylhydrazono)]-3,3-dimethyl-2-butanone (3): 4-Chloroaniline (18.92 g 148 mmol) was dissolved in 100 mL of 6 N HCl and 44 mL of water and placed in an ice bath with stirring for 30 min to give a gray suspension. Sodium nitrite (11.2 g 162 mmol) which had been dissolved in 60 mL of water was added dropwise to the aniline. The diazotization reaction stirred on ice for 1 h. Separately, 3-chloro-5,5-dimethyl-2,4-dihexanone (2) (26.1 g 148 mmol) was dissolved in 200 mL of pyridine–water (1:1) and cooled on ice. The diazo salt formed above was added dropwise to the chloroform solution. Upon addition, the ice bath was removed and the reaction was allowed to warm to room temperature with hard stirring for 2 h. The reaction mixture was extracted with two 250 mL portions of ethyl acetate, the ethyl acetate fractions were then washed twice with 500 mL portions of 2 N HCl and once with 300 mL of brine. The organic fraction was dried over sodium sulfate and the solvent was evaporated under reduced pressure to give a reddish paste. The crude material was suspended in 150 mL of methanol where it dissolved completely upon heating to reflux for 1 h. The solution was cooled to 4 °C overnight giving a precipitate. The precipitate was recovered by filtration, washed with hexanes, and dried under high vacuum to give 22.6 g (83 mmol, 56%) of orange solid (3). LC/MS $M/Z$ = 271.1 (M−H)− (calculated), $M/Z$ = 271.1 (found). This material was not further purified or characterized by NMR but was taken directly into the next step.

TY-52156: [1-Chloro-1-(4-chlorophenylhydrazono)]-3,3-dimethyl-2-butanone (3) (10 g 36.6 mmol) and 4-chloroaniline (5.14 g, 40.3 mmol) were dissolved in 125 mL of ethanol and cooled on an ice bath with stirring. Triethylamine (4.45 g, 43.9 mmol) was added to the cold solution and the ice bath was removed. The reaction was allowed to warm to room temperature while stirring for 3 h. The solvent was removed under vacuum to give a paste which was then dissolved in 50 mL of water. The aqueous solution was extracted into two 100 mL portions of ethyl acetate and the combined organic phases were washed with 50 mL of brine and then dried over sodium sulfate. The solvent was removed under vacuum. The resulting crude material was purified using a 120 g silica gel cartridge eluting with a gradient of 5–10% ethyl acetate in hexanes over 16 column volumes. Product fractions were combined and concentrated under vacuum to give TY-52156 as a light yellow foam (9.5 g, 26.1 mmol, 71%). $^1$H NMR (400 MHz, CDCl3) δ 7.16–7.29 (m, 5H), 6.99 (d, $J$ = 8.8 Hz, 2 H), 6.76 (s, 1H), 6.55 (d, $J$ = 8.7 Hz, 2H), 1.47 (s,

9H); $^{13}$C NMR (101 MHz, CDCl3) δ 200.8, 141.8, 138.3, 133.7, 129.6, 129.4, 127.0, 126.9, 118.5, 115.1, 43.4, 28.6; LC/MS $M/Z$ = 364.1 (M+H)+ (calculated), $M/Z$ = 364.2 (found). The synthetic route to TY-52156 is shown in Figure 10.

**TY-52156 efficacy experiments**. Five to seven-week-old athymic NIH nu/nu mice were anesthetized with isoflurane/O$_2$ and injected in the left cardiac ventricle with 231-BR cells ($1.75 \times 10^5$ cells in 0.1 mL Ca$^{2+}$ Mg$^{2+}$ free PBS). At day 24 post-injection for the 231-BR model, mice were randomized (based on body weights using StudyLog software randomization feature) to vehicle or 10 mg kg$^{-1}$ of TY-52156 (in 0.5 w per v% Tragacanth Gum (MP Biomedicals, Inc.)) groups, BID by oral gavage. The treatments were administered for 4 days. On day 28 for the 231-BR model, mice received TRD followed by a perfusion as described above. It was expected that some mice would require euthanasia before the end of the treatment because of neurological or morbid symptoms (per ACUC rules): those mice were excluded from the data analysis. At necropsy, brains were fresh frozen in OCT. Ten series of five 8 μm-slices (2 slices per slide) were sectioned every 300 μm, through the entire brain in a ventral to dorsal orientation. The first section in each series of 5 was H&E stained to localize and count the metastasis clusters, and the following slides were used to photograph TRD exudation and IF with the following markers: CD31 (1:500—cat#550274, BD Pharmingen), human cytokeratin (1:100—cat# M0821, DAKO) and Ki67 (1:100—cat# M7240, Dako). Photographs were acquired and analyzed by the Zeiss software, ZEN software. For the 231-BR model, the vehicle group contained 11 mice and the TY-52156 treated group had eight mice.

TRD diffusion was quantified on a continuous rather than dichotomized basis, measuring the surface area covered by TRD. For each mouse, one section on one slide (two tissue sections per slide) from the series with the highest number of metastases was used for TRD quantification. The image was captured at 10×, over 5–10 s interval, to avoid thawing the tissue. This same slide was used for IF staining: Ki67, cytokeratin and CD31 on one section, and S1P3, GFAP and CD31 on the other. ZEN software enabled identification of the exact coordinates of the pictures taken for TRD. After IF staining, the same coordinates were used to capture images of the same cells, enabling normalization of TRD to cytokeratin staining as a marker of metastases burden, or to CD31 as a marker of blood vessel density. For each IF marker, percent of area over the entire picture was measured. Two mice per group did not receive TRD, for PK analysis. Data are representative of two experiments performed.

A similar experiment was performed using the SUM190-BR model. Five to seven-week-old athymic NIH nu/nu mice were anesthetized with isoflurane/O$_2$ and injected in the left cardiac ventricle with SUM190-BR ($5 \times 10^5$ cells in 0.1 mL Ca$^{2+}$ Mg$^{2+}$ free PBS). At day 47 post-injection for the SUM190-BR model, mice were randomized to vehicle or 10 mg kg$^{-1}$ of TY-52156 (in 0.5 w per v% Tragacanth Gum) groups, b.i.d. by oral gavage. The treatments were administrated for 4 days. On day 50 for the SUM190-BR model, mice received TRD followed by a perfusion as described above. Several mice had to be euthanized before the end of the treatment. The vehicle group had five mice and the TY-52156 treated group had eight mice. At necropsy, the brains were frozen in OCT, sectioned, staining with H&E (to count the metastases), and immune-stained with cytokeratin and CD31 (to normalize TRD to metastasis area and blood vessel density).

**Pharmacokinetics of TY-52156**. For two mice per group of the TY-52156 experiment, terminal bleeds and the dissected perfused brains were analyzed. Blood was processed to plasma, precipitated with acetonitrile and centrifuged before the supernatant was analyzed via LC-MS/MS for TY-52156 measurement. Brains were rinsed with saline and snap frozen until analysis, when they were homogenized with water (100 mg mL$^{-1}$ tissue concentration), precipitated with acetonitrile, and the supernatant analyzed by LC-MS/MS. Briefly, a Waters AQCUITY BEH® C18 (2.1 × 50 mm, 1.7 μm) column was used to isolate TY-52156 before identification via the mass transition $M/Z$ 362.2 → 209.8. Assay calibration ranges were 0.5–5000 ng mL$^{-1}$ for plasma; 10–5000 pg mg$^{-1}$ for tissue.

**Pharmacokinetics of methotrexate**. Following a 50 mg kg$^{-1}$ intraperitoneal dose of MTX in mice bearing 231-BR metastases, blood and whole brains were obtained, with the brains rinsed with saline and snap frozen. Blood was processed to plasma, then 50 μL of plasma from each sample was mixed with 3× volume of methanol (containing 1 μg mL$^{-1}$ sorafenib as internal standard) to precipitate plasma proteins. Whole brains were weighed, then 10 μL water per mg of tissue was added to homogenize. The tissue homogenate was treated same as plasma from here on. This mixture (sample matrix in 3× methanol) was vortexed and centrifuged, and the resulting supernatant injected onto a Waters ACQUITY UPLC BEH C18 column (2.1 × 50 mm, 1.7 μm). Column eluent was directed into a mass spectrometer to detection based on the positive ion mass transition of $M/Z$ 455.4 → 308.1; sorafenib was detected based on the positive ion mass transition of $M/Z$ 465.0 → 252.0. The assay had a calibrated range of 10–5000 ng mL$^{-1}$ for plasma, and brain 50–50,000 pg mg$^{-1}$.

Fig. 10 caption:

**Fig. 10** Synthetic route to TY-52156. Reagents and conditions: **a** potassium *t*-butoxide, diisopropyl ether; **b** sulfuryl chloride, CHCl$_3$; **c** HCl, water, sodium nitrite; **d** pyridine–water; **e** 4-chloroaniline, ethanol, triethylamine

**Tumor cell overexpression of S1P3**. MDA-MB-231-BR cells were transduced with lentiviruses for S1P3 or control vector (GeneCopoeia, LPP-U1124-Lv152–200 or EX-EGFP-Lv152) along with 5 μg mL$^{-1}$ polybrene. Cells were selected with 300 μg mL$^{-1}$ Hygromycin B. S1P3 expression was confirmed with RT-PCR and IF.

The 231-BR-Vector and the 231-BR-S1P3 lines ($1.75 \times 10^5$ cells in 0.1 mL $Ca^{2+}$ $Mg^{2+}$ free PBS) were injected in the left cardiac ventricle of 5–7-week-old athymic NIH nu/nu mice, under anesthesia with isoflurane/$O_2$. At the endpoint, mice received TRD followed by a perfusion as described above. Subsequently, brains were fresh frozen in OCT and sectioned and stained as described above. The 231-BR-Vector injected group contained 12 mice and the 231-BR-S1P3 group contained 11 mice. S1P3 overexpression increased the number of micrometastases. Therefore, to study TRD diffusion among mice with comparable metastatic burden, mice whose brains had less than 150 micrometastases in 231-BR-Vector injected group ($n = 2$) or had >600 micrometastases in the 231-BR-S1P3 group ($n = 2$) were excluded from the TRD study.

**Patients' brain metastasis specimens.** Human craniotomy specimens were collected by the Massachusetts General Hospital Cancer Center, Harvard Medical School (IRB 10–454), the Military Institute of Medicine and the Copernicus Hospital Gdańsk in Poland (Bioethics committee approval number: 52/WIM/2015), and the Hospital Clairval, and stored in the AP-HM tumor bank (AC-2013-1786), in France. Informed consents were obtained from all the subjects for the samples coming from France and coming from Massachusetts General Hospital Cancer Center. Informed consents were waived for the other samples, as the patients were deceased. All the samples, regardless of the country of origin, were anonymized and approved by the Office of Human Subjects Research Protections (OHSRP) at the National Institutes of Health (OHSRP #13093). IF staining was performed as described previously[19]. Briefly, tissues were fresh frozen and then embedded in OCT. H&E staining and IF with S1P3 and GFAP antibodies (sources above) were performed on 8 μm sections. Pictures were acquired with Zeiss Axioskop and ZEN software[19]. Another cohort of patients' brain metastasis samples was immune-stained and analyzed by pathologists from the Medical University of Gdańsk, Poland.

**Generation of immortalized brain-derived cell lines.** Primary human astrocytes (#1800) and brain vascular pericytes (#1200) were purchased and cultured in astrocyte medium (#1801) and pericyte medium (#1201), respectively, according to manufacturer's protocol (ScienCell research laboratories). Primary human brain microvascular endothelial cells (ACBRI 376) were purchased and cultured in CSC complete medium (4Z0-500) (Cell Systems Corporation). Lentiviruses coding for SV40 Large T antigen were purchased (G256, Applied Biological Materials Inc) and transduced along with 5 μg mL$^{-1}$ polybrene (Sigma-Aldrich) for 6 h and later drug selected with 1 μM puromycin (Sigma-Aldrich) to give immortalized human astrocytes (HAL), endothelial cells (HEL) and pericytes (HPL). Immunostaining for GFAP, ZO-1, and NG2 was performed to confirm phenotypes of the cells (Supplementary Fig. 5B). Standard procedures were used for Western blot analysis.

**In vitro BBB and BTB assays.** $1 \times 10^3$ immortalized pericytes (HPLs) in pericyte medium (PM) were seeded on abluminal side of 0.4 μm transwell insert (Corning, #353095) kept inverted in a 6-well plate. After 6 h, the luminal side of inserts were coated with attachment factor (Cell Systems Corporation, #4Z0-210) and $5 \times 10^4$ immortalized endothelial (HEL) cells per insert were seeded. After 3 days of culture, $5 \times 10^4$ immortalized astrocytes (HALs) in astrocyte medium (AM) were seeded in a 24-well plate. After the cells attached, the culture media was replaced with PM, and the inserts with HEL and HPL were placed in the wells. Various assays were started the next day which was considered time 0 for treatment. The following antagonists were used in the study: EX26 (PMID: 23204443) (#SML1680, Sigma Aldrich), an S1P1 inhibitor, at 1 nM; JTE-013 (PMID: 12445827) (#J4080, Sigma Aldrich), an S1P2 inhibitor, at 17 μM; TY-52156 (PMID: 20097776), an S1P3 inhibitor, at 2 μM; CAY10444 (PMID: 12361389, 23723371, 28300348, 27282481) (#10005033, Cayman chemical), another S1P3 inhibitor, at 10 μM; and CYM50358 (PMID: 21570287) (#SML1066, Sigma Aldrich), an S1P4 inhibitor, at 25 nM. Cells were treated for 4 h and TEER was measured after 24 h. In case of BTB, 1-week-old tumor spheres formed by 231-BR cells were seeded on to the astrocytes before time 0. TEER was measured using EVOM2, Volt/Ohm Meter (World Precision Instruments). At the indicated time point, a chopstick electrode (STX2 or STX3), mounted on a stand to avoid fluctuations, was used to measure resistance. Three readings were taken per well, and coated insert without any cells was used as negative control. The value of the negative control was subtracted from test values, and these normalized TEER values were multiplied by 0.33 to obtain Ω cm$^{-2}$. These values were used to calculate fold changes.

At the end of the experiment, the inserts were washed once with PBS and fixed with 4% PFA at room temperature for 15 min. The immunostaining was performed as described above using ZO-1 (1:100, Thermo Scientific, #61-7300), and VE-cadherin (1:100, Abcam, #ab33168) antibodies. The membranes were cut using a scalpel, trimmed with a scissor, placed on a glass slide, mounted with fluorescent mounting medium (Dako, # S3023) and imaged.

**Permeability assay.** For assessing the permeability of these in vitro BBB/BTB, after TEER measurement, the transwell inserts with endothelial cells and pericytes were placed in a new 24-well plate containing 1 mL PBS. Endothelial media containing 100 μg mL$^{-1}$ Doxorubicin (Ex/Em:480/590 nm) were added on top of endothelial cells in the luminal side. For assessing the passage of these fluorescent

molecules, which is directly proportional to the permeability of the BBB/BTB, 100 μL PBS was collected after 45 min incubation into 96-well clear bottom plates (Thermo Scientific, #165305). Fluorescence (at 480/590 nm) was measured in SpectraMax M2 (Molecular Devices). PBS was used as negative control and coated insert without cells was used as a positive control.

**S1P3 knockdown.** For studying the specific role of S1P3 in astrocytes, S1P3 mRNA expression was stably knocked down using lentiviruses (sh*S1PR3*-1 from Sigma Aldrich, #SHCLNV-NM_005226, sh*S1PR3*-2 from Applied Biological Materials Inc, #i021637c) coding shRNA targeted against 2 different regions of *S1PR3* (sh*S1PR3*-1: CCGGGCGGCACTTGACAATGATCAACTCGAGTTGATC-AT TGTCAAGTGCCGCTTTTT; sh*S1PR3*-2: CGCATCTACTTCCTGGTGAAGTC CAGCAG). The knockdown was confirmed using RT-PCR and IF (Fig. 6a). Control vectors were MISSION® pLKO.1-puro Non-Target shRNA control transduction particles (#SHC016V, Sigma Aldrich) and Scrambled siRNA GFP lentivector (#LV015G, Applied Biological Materials).

**Cytokine profiling.** Assessment of cytokines secreted by shControl and *S1PR3*-knocked down HAL cells was performed using Proteome profiler- human cytokine array (R&D Systems, #ARY005B). Media from ~70% confluent HAL-shControl-1,2 and HAL-sh*S1PR3*-1,2 was replaced with serum-free HA medium. 12–16 h later, the media was harvested, particulates were removed by brief centrifugation, aliquoted and frozen. Fresh aliquots were quantified, and ~500 μL was used for the assay according to the manufacturer's protocol. Using ImageJ software, the intensity of quantifiable spots was measured. The intensities were normalized using reference spots from both membranes. Fold change of cytokine expression in sh*S1PR3* astrocytes compared to shControl cells was calculated and graph was plotted.

**Cytokine inhibition.** Once the in vitro BBB/BTB cultures were established as described above, culture media on astrocytes was changed to serum-free AM. TEER was measured (0 h) and media containing the following neutralizing antibodies were added onto astrocytes in respective wells: (from R&D systems): 1 μg mL$^{-1}$ IL-6 (MAB206), 1 μg mL$^{-1}$ IL-8 (MAB208), 2 μg mL$^{-1}$ CCL2 (MAB679), 3 μg mL$^{-1}$ CXCL1 (MAB276), 1 μg mL$^{-1}$ GM-CSF (MAB215), and isotype rabbit IgG (X0944 —Dako). TEER was measured after 0 and 24 h. These inserts were used later for permeability assays and immunostaining.

**CCL2 ELISA.** Immortalized astrocytes (HALs) were cultured in complete medium. When the cells were approximately 70% confluent the media was discarded, cells were washed twice with PBS and fresh serum-free media (AM) containing DMSO or various antagonists (EX26: 1 nM, JTE-013: 17 μM, TY-52156: 2 μM, CAY10444: 10 μM, and CYM50358: 25 nM) were added. After 4 h media was discarded, washed twice with PBS and fresh serum-free media was added. 24 h post treatment, the culture supernatant was collected, centrifuged to remove cell debris and used to detect CCL2 using Human MCP-1/CCL2 ELISA MAX™ Deluxe Set (# 438804, Biolegend) according to manufacturer's protocol. For knockdown studies, shControl and sh*S1PR3*-1, 2 cells were grown in complete medium, when cells were 70% confluent, media was changed to serum-free conditions and culture supernatants were harvested and assayed after 24 h. The relative CCL2 levels were plotted as normalized absorbance at 450 nm.

**Cell growth assay.** Cells were seeded at 30,000 cells per well, in 96-well plates. The next day, the cells were treated with DMSO or various antagonists (EX26: 1 nM, JTE-013: 17 μM, TY-52156: 2 μM, CAY10444: 10 μM, and CYM50358: 25 nM) for 4 h. 24 h post treatment, the cell growth was measured using AlamarBlue® Cell Viability Assay (#DAL1100, ThermoFisher) according to manufacturer's protocol. Percent cell growth was calculated using normalized fluorescence at 570 nm/585 nm and plotted.

**Statistical analysis.** Power analysis indicated that a minimum of $n = 4$ was required to detect a significant small-to-medium effect size, with an acceptable power of 0.8. For in vitro data, four to seven technical replicates of each experiment were performed. For each replicate, three measurements were recorded at each time point. When $n$ was less than 5, individual data points were plotted in the graph. For in vivo data, power calculation indicated that 8–11 mice per group were required to obtain a statistical power of 0.8. An exception is Fig. 7d, e, for which only five mice were used in the vehicle control (several mice had to be euthanized earlier because of morbid symptoms).

All the data were analyzed using the software Graphpad Prism 7.01 (GraphPad Software, Inc.). Nonparametric tests were performed on datasets not normally distributed. Specific tests are mentioned in each figure legend. Briefly, for bivariable analysis, two-tailed Mann–Whitney test was chosen for unpaired data and Wilcoxon rank test was chosen for matched-paired data. For multivariable analysis, a non-parametric Levene's test was performed with SPSS software from IBM to verify the equality of variances in the samples ($P > 0.05$). Subsequently, Kruskal–Wallis test followed by Dunn's multiple comparison test was performed.

To analyze interaction between S1P3 overexpression and TRD diffusion, Chi-square statistical analysis was performed. We used $P < 0.05$ for significance.

**Data availability**. The authors declare that all the data supporting the findings of this study are available within the paper and its supplementary information files. Full gels and Western blots are presented in Supplementary Fig. 12. The microarray data discussed in this publication have been deposited in NCBI's Gene Expression Omnibus and are accessible through GSE112776 [https://www.ncbi.nlm.nih.gov/geo/query/acc.cgi?acc=GSE112776].

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

## Acknowledgments

The authors acknowledge the statistical advice of Dr. Seth Steinberg and Mr. David Liewher, Biostatistics and Data Management Section, CCR, NCI, Bethesda, MD, USA. This work was supported by the Intramural program of the National Cancer Institute, U. S. Department of Defense Breast Cancer Research Program, grant number: W81 XWH-062-0033, and a research grant from the Inflammatory Breast Cancer Foundation. The human tissue samples from France were provided by AP-HM tumor bank AC-2013-1786, BB-0033-00097. The immunohistochemistry of human tissue samples performed in Poland was partly funded by the Institutional Grant ST-23 from the Medical University of Gdańsk. This project has been funded in whole or in part with Federal funds from the National Cancer Institute, National Institutes of Health, under Contract No. HHSN261200800001E. The content of this publication does not necessarily reflect the views or policies of the Department of Health and Human Services, nor does mention of trade names, commercial products, or organizations imply endorsement by the U.S. Government. Frederick National Laboratory is accredited by AAALAC International and follows the Public Health Service Policy for the Care and Use of Laboratory Animals. Animal care was provided in accordance with the procedures outlined in the "Guide for Care and Use of Laboratory Animals" (National Research Council; 2011; National Academies Press; Washington, D.C., USA).

## Author contributions

B.G. and A.N.P. conceptualized, designed, performed, and analyzed the experiments. P.S. S. coordinated and directed the project, conceptualized and designed experiments, and wrote the manuscript. S.W. analyzed the microarray data. X.W. performed the microarray experiments. J.H. helped in the development of the LCM methodology. C.R. and S. D. performed the mouse experiments until necropsy. E.H. and E.L.D. processed mouse tissues. C.J.P., O.M.H., and W.D.F. performed the pharmacodynamic analysis of the TY-52156. G.T.P. and J.P.S. synthetized TY-52156 compound. R.D., E.I-S., W.B, R.P., J.J., W. K., P.K.B., N.N., E.B., and P.M. provided the clinical materials (including interaction with patients and preparation of human specimen blocks).

## Additional information

**Competing interests:** The authors declare the following competing interests: P.K. Brastianos reports receiving speaker's bureau honoraria from Genentech and Merck, and is a consultant/advisory board member for Genentech. P.S. Steeg reports receiving a commercial research grant from Genentech and Medimmune. R. Duchnowska reports consulting personal fees from Roche, GSK, Novartis, Lilly, Pfizer, AstraZeneca, Amgen, Boehringer Ingelheim, Teva, and Egis. The other authors declare no competing interests.

