## [Peer Review File · Nature Communications]

Reviewers' comments:

Reviewer #1 Expert in brain metastases:

This manuscript by Gril and co-workers deals with a very important topic, highly relevant for translational concepts: what is the nature of the blood-tumor barrier in and around brain metastases, and how can delivery of therapeutics be improved? Following a

The manuscript is very well written, the experiments appear to be carefully performed, the results are interpreted in a valid way, and the discussion is very good. The BBB model, in its complexity, seems a good model to answer some of the important questions.

I have to major issues:

1. It would be important to get deeper knowledge about how astrocyte S1P3 modulates the blood-tumor barrier in brain mets: how much is an effect on astrocytes themselves, how much on ECs (paracrine effect), etc. The letter can be nicely addressed in the BBB model, and then e.g. verified in histology sections.

2. It would greatly increase the value of this study to get a clue (or at least a hint) whether interference with S1P3 could increase the antitumor activity of systemic chemotherapeutics (or targeted drugs) on brain mets. I understand that S1P3 activation is difficult to achieve pharmacologically. What about overexpressing S1P3 in brain metastasizing cancer cells, and see whether this will increase drug uptake and anti-metastasis activity? Alternatively, the authors might consider to use transgenic mice that are available where the sphingosine pathway is overactive. Even though this would also not translate into a direct translational benefit, it would add very useful information to the important question whether interference with BBB permeability is (or is not) helpful for brain metastasis therapy in living organisms.

Minor points:

Figure 3: the IF histology images are difficult to assess, one would need a higher mag close-up, comparable to what the authors have done in Figure 2

The introduction sometimes reads like a discussion, and is too lengthy. I would reduce the introduction significantly, and move many aspects to the discussion section. There is no need to introduce SP1, since this is the result of a screening, and not an a priori starting point of this study.

Abstract: "...in human craniotomies": wording is somewhat misleading; please change to "patients' brain metastases" or similar.

Reviewer #2 Expert in metastasis:

The authors have addressed the clinically important problem of poor penetration of systemic chemotherapy penetration into brain metastasis. This group has previously demonstrated that the brain tumor barrier "BTB" is profoundly heterogeneous, with some brain metastasis-associated vasculature more porous than others. This manuscript seeks to address the mechanism behind this heterogeneous BTB. The authors employ a single immunocompromised model, MDA 231-BR, to generate brain metastases (both high and low permeability BTBs). Gene expression profiling of these brain metastases and their associated microenvironment uncovered elevated S1P1 gene expression in the microenvironment. Immunofluorescence and in vitro BTB models suggested that

astrocyte S1P1 is responsible for this alteration in BTB. Inhibition of signaling from this GPCR with a compound, TY-52156, alters BTB permeability in vivo. Despite the importance of this question, the experimental design and analyses limit the interpretation of the results presented and preclude recommendation of this manuscript for the broad readership of Nature Communications.

Major Points:

1. The initial laser-capture microdissection followed by gene expression profiling was undertaken from a single mouse model, MDA 231-BR. This discovery method drives all of the subsequent work in this manuscript. The authors do employ two additional models as validation tools J1MT-1-BR and SUM190-BR; it is clear that both low permeability and high permeability brain metastases may be isolated from these other models. The laser-capture microdissection and RNASeq should be undertaken in at least two models, if not all three. Genes differentially expressed in all three models are then prime candidates for further mechanistic study.

2. The low number of clinical samples of brain metastases is unfortunate. These data provide the only clinical validation present in this manuscript. If additional samples cannot be secured for analysis, quantitative, objective analysis of these sparse data is essential. The current subjective scoring system (Supp Table 3) and representative images (Figure 4) are inadequate. One option is to quantitate the proportion of S1P3 (+) astrocytes per section. Alternatively, automated image analysis such as metamorph might be employed to assess the proportion of GFAP (+) area that is also S1P3 (+).

3. Pharmacologic experiments performed are incomplete. The use of a single S1P3 antagonist (TY-52156) is not ideal. If the authors are unable to employ an additional antagonist, they should address the specificity of this compound, addition of S1P (the ligand) to abrogate drug effect, (such as in Figure 5B-E).

4. Given the importance that the authors place on the inflammatory response, the reliance on immune-deficient models strongly limits interpretation of these results. If the authors wish to continue this line of reasoning, they will need to expand their models to include a syngeneic model. This will enable the use of a S1P3 knockout mouse as host.

Minor Points:

1. Immunofluorescence of normal brain and poorly permeable lesions should be presented alongside that from the highly permeable lesions in Figure 3A. This is described in the text but not shown. This might be accomplished with a low-power image containing both the brain metastasis and surrounding adjacent normal brain parenchyma.

2. The knock-down experiments (Figure 6) employ a single short hairpin to knock-down S1P3. To address the possibility of off target effects, these must be repeated with an additional independent short hairpin. Alternatively, CRISPRi might be employed for the same purpose.

3. Considerable work by other laboratories has taken place regarding astrocyte-brain metastatic cancer cell interactions. The authors should address the work by Kim SJ et al 2011 Neoplasia, as the datasets obtained after in vitro co-culture of MDA231 and astrocyte co-culture bear similarities to the data presented in this manuscript.

4. The manuscript is poorly prepared. Writing throughout the manuscript is informal, and filled with colloquialisms that are not readily understood by readers who do not speak English as a first language: e.g. line 60 "at levels higher than normal brain, but with peak levels ~a log less than"

5. References are often absent or inadequate: e.g. line 70 "Interestingly, in two preclinical model systems, another 7% and 22-49% of metastases demonstrated >30 and >10 fold paclitaxel uptake about the BBB, respectively" (no reference given).

Response to Reviewers

We thank the reviewers for their insightful comments, they have truly improved the manuscript. Please find each comment listed below and our response.

Almost every figure of the revised manuscript has been improved. Salient changes include:

- Inclusion of an immunocompetent brain metastasis model for S1P3 expression analysis
- Inclusion of additional frozen human craniotomy specimens for S1P3 staining
- Development of a quantification system for S1P3 expression for the craniotomy specimen staining
- Inclusion of a second S1P3 antagonist in BBB/BTB *in vitro* assays
- Inclusion of a panel of antagonists to other S1P receptor family members in the *in vitro* BBB/BTB assays, to demonstrate the specificity of S1P3
- Inclusion of a second, independent shRNA to S1P3 in BBB/BTB *in vitro* assays
- Addition of a second model system for the *in vivo* effect of TY-52156 on BTB permeability
- Development of an experimental schema to quantify drug uptake in the brain in response to TY-52156 *in vivo*
- Validation of S1P3 regulation of astrocytic chemokine secretion using two shRNAs and the panel of antagonists
- Transfection of brain metastasizing tumor cells with S1P3 and evaluation of BTB permeability.

The manuscript as revised presents novel data indicating that the blood-tumor barrier of brain metastases may be a tractable molecular target for improving drug efficacy.

Reviewer #1 Expert in brain metastases:

This manuscript by Gril and co-workers deals with a very important topic, highly relevant for translational concepts: what is the nature of the blood-tumor barrier in and around brain metastases, and how can delivery of therapeutics be improved?

The manuscript is very well written, the experiments appear to be carefully performed, the results are interpreted in a valid way, and the discussion is very good. The BBB model, in its complexity, seems a good model to answer some of the important questions.

Response: We thank the reviewer for emphasizing the importance of our research.

I have two major issues:

1. It would be important to get deeper knowledge about how astrocyte S1P3 modulates the blood-tumor barrier in brain mets: how much is an effect on astrocytes themselves, how much on ECs (paracrine effect), etc. The letter can be nicely addressed in the BBB model, and then e.g. verified in histology sections.

Response: The reviewer identifies an important point. In the original submission we found by staining that S1P3 overexpression was principally present in activated astrocytes in the reactive neural microenvironment. Multiple lines of evidence indicate that astrocytes are the causal cell type: removal of astrocytes from the *in vitro* BBB/BTB assays abrogates the effect of the S1P3 inhibitor TY-52156 (Fig 5e in the original submission, current Fig 5h, in which we tested the second S1P3 antagonist). Using shRNAs, diminution of astrocytic S1P3 significantly altered BBB/BTB permeability *in vitro* (Fig. 6 in the original submission, now two independent shRNAs in Fig. 6). Endothelial S1P3 has been reported, however it was not at levels detectable by immunofluorescence in either mouse brains or human craniotomies in our studies.

The presence of S1P3 in activated astrocytes was verified in human brain metastasis specimens (original submission, Fig.4). To strengthen this observation, the initial cohort of fresh frozen patients' samples was doubled, and quantification of staining was performed (revised Supplementary Table 3 and Supplementary Fig. 7).

We have delineated a downstream pathway using the *in vitro* BBB/BTB assays that involves S1P3 alteration of astrocyte secretion of IL-6 and CCL2, both of which alter BBB permeability *in vitro* and endothelial adhesion and tight junction protein expression (current Fig. 8 and supplementary Fig. 11). Thus, both cell types are involved, one downstream of the other. Staining for other elements of this pathway is precluded by lack of tissue availability, only small pieces of fresh frozen craniotomy specimens were made available.

2. It would greatly increase the value of this study to get a clue (or at least a hint) whether interference with S1P3 could increase the antitumor activity of systemic chemotherapeutics (or targeted drugs) on brain mets. I understand that S1P3 activation is difficult to achieve pharmacologically. What about overexpressing S1P3 in brain metastasizing cancer cells, and see whether this will increase drug uptake and anti-metastasis activity? Alternatively, the authors might consider to use transgenic mice that are available where the sphingosine pathway is overactive. Even though this would also not translate into a direct translational benefit, it would add very useful information to the important question whether interference with BBB permeability is (or is not) helpful for brain metastasis therapy in living organisms.

Response:

This comment addresses several issues: (1) can we measure drug uptake instead of Texas red dextran to evaluate permeability; (2) Will S1P3 overexpression in brain metastasizing tumor cells achieve the desired result? (3) Can transgenic mice be used to confirm this pathway? The reviewer is correct that S1P3 activation is presently difficult. A literature search was conducted in our lab and an expert in the field (Dr. Richard Proia) was consulted. He identified CYM 5541, but it is unacceptable: it is not brain permeable, and it induces cardiac arrest. (<http://dx.doi.org/10.1124/mol.115.100222>).

To address the first aspect, we designed a new mouse experiment in collaboration with Drs. Doug Figg and Cody Peer, both in Clinical Pharmacology at the NCI. Drs. Figg and Peer chose methotrexate (MTX) for the experiments for three reasons: 1- Methotrexate can moderately cross the brain-blood barrier, so a decrease in drug uptake could potentially be observed, 2- it is a weak substrate for ABC transporters, reducing the risk of drug efflux as a confounding variable, 3- Drs. Figg and Peer have a validated assay to measure MTX concentration via liquid chromatography tandem mass chromatography (LC–MS/MS).

Using the 231-BR model, mice received TY-52156 or vehicle for the last four days when brain metastases were already established, then then two doses of MTX. Drs. Figg and Peer measured the concentration of MTX in the plasma and in the whole brain lysate of each animal. The ratio [MTX] brain/plasma is shown in a new Supplementary Fig. 9. We observed a trend of [MTX] brain/plasma reduction the S1P3 antagonist group. We consider these data very encouraging as the Texas Red dextran (TRD) and MTX assays were completely different: for TRD, drug uptake in brain metastatic lesions was quantified. For MTX, an entire brain, which is most uninvolved, was homogenized. In our opinion, the data provide quantifiable evidence of an *in vivo* effect of TY-52156 on drug uptake.

We congratulate Reviewer 1 for conceiving the next experiment. He/she asked if tumor cells could carry S1P3 behind the BTB, signal as per reactive astrocytes, and affect permeability? To our surprise, S1P3 overexpressing 231-BR cells showed increased IL-6 and CCL2 secretion *in vitro* in the same manner as reactive astrocytes. Injection of the S1P3 overexpressing cells produced metastases with the typical spread of permeabilities, but significantly greater metastases were above minimal permeability (Fig. 9).

A transgenic mouse model could help confirm and extend our observations. Currently we have no syngeneic brain metastasis model for the *S1PR3* mice available- the 4T1 assay is in a different strain.

Minor points:

Figure 3: the IF histology images are difficult to assess, one would need a higher mag close-up, comparable to what the authors have done in Figure 2

Response: A higher magnification window was added on each IF picture.

The introduction sometimes reads like a discussion, and is too lengthy. I would reduce the introduction significantly, and move many aspects to the discussion section. There is no need to introduce SP1, since this is the result of a screening, and not an a priori starting point of this study.

Response: Text has been edited.

Abstract: "..in human craniotomies": wording is somewhat misleading; please change to "patients' brain metastases" or similar.

Response: the wording was changed to "patients' brain metastases".

Reviewer #2 Expert in metastasis:

The authors have addressed the clinically important problem of poor penetration of systemic chemotherapy penetration into brain metastasis. This group has previously demonstrated that the brain tumor barrier "BTB" is profoundly heterogeneous, with some brain metastasis-associated vasculature more porous than others. This manuscript seeks to address the mechanism behind this heterogeneous BTB. The authors employ a single immunocompromised model, MDA 231-BR, to generate brain metastases (both high and low permeability BTBs). Gene expression profiling of these brain metastases and their associated microenvironment uncovered elevated S1P1 gene expression in the microenvironment. Immunofluorescence and in vitro BTB models suggested that astrocyte S1P1 is responsible for this alteration in BTB. Inhibition of signaling from this GPCR with a compound, TY-52156, alters BTB permeability in vivo. Despite the importance of this question, the experimental design and analyses limit the interpretation of the results presented and preclude recommendation of this manuscript for the broad readership of Nature Communications.

Major Points:

1. The initial laser-capture microdissection followed by gene expression profiling was undertaken from a single mouse model, MDA 231-BR. This discovery method drives all of the subsequent work in this manuscript. The authors do employ two additional models as

validation tools JIMT-1-BR and SUM190-BR; it is clear that both low permeability and high permeability brain metastases may be isolated from these other models. The laser-capture microdissection and RNASeq should be undertaken in at least two models, if not all three. Genes differentially expressed in all three models are then prime candidates for further mechanistic study.

Response: The laser capture microdissection technique was combined with a brain metastasis assay, gene expression profiling and validation at the protein level. This setup required a year of pilot experiments to detect the permeable vs. impermeable metastases, collect good quality RNA from microdissected tissue, and generate the heatmap. As communicated to the editor in a previous email, the generation of another heatmap is beyond the scope of the present manuscript. Most importantly, several candidate genes picked from the heat map (S1P3, 2 GABA subunits and EphrinA5) were validated in two independent model systems at the protein level, indicating that this is a good dataset. The readership can interrogate many other genes listed in other experiments. The S1P3 trend has now been validated in an additional immune competent model.

2. The low number of clinical samples of brain metastases is unfortunate. These data provide the only clinical validation present in this manuscript. If additional samples cannot be secured for analysis, quantitative, objective analysis of these sparse data is essential. The current subjective scoring system (Supp Table 3) and representative images (Figure 4) are inadequate. One option is to quantitate the proportion of S1P3 (+) astrocytes per section. Alternatively, automated image analysis such as metamorph might be employed to assess the proportion of GFAP (+) area that is also S1P3 (+).

Response: We agree with the Reviewer. Extensive efforts were made to strengthen the clinical data by both increasing the sample cohort and quantifying the staining. We have contacted neuro-oncologists in France, from the Cleveland Clinic, at NCI and elsewhere looking for additional frozen samples. Of those available, not all were properly preserved. In our original submission, 9 samples were presented. The revised manuscript has more than doubled this amount to 19 brain metastasis samples.

These samples were analyzed using a newly developed quantification protocol as suggested by the Reviewer. Using Zen 2, Zeiss the proportion of GFAP+ astrocytes that were S1P3+ was quantified. These data are presented in Supplementary Table 3 in the revised paper. In addition, our colleagues in Poland worked extensively on a staining protocol in formalin-fixed, paraffin embedded (FFPE) material. S1P3 staining on paraffin-embedded tissues has been previously published by Van Door et al. 2010 (DOI 10.1002/glia.21021). Unfortunately, the S1P3 antibody used in this publication is not commercially available anymore. Therefore, our

colleagues had to optimize a new protocol. Subsequently, a pathologist analyzed brain metastasis samples from 31 patients (revised Supplementary Fig. 6). This second validation is noteworthy as it confirms the presence of astrocytic S1P3 in human samples by an independent team and a different cohort of patients.

3. Pharmacologic experiments performed are incomplete. The use of a single S1P3 antagonist (TY-52156) is not ideal. If the authors are unable to employ an additional antagonist, they should address the specificity of this compound, addition of S1P (the ligand) to abrogate drug effect, (such as in Figure 5B-E).

Response: The Reviewer makes an important point. An additional S1P3 antagonist (CAY10444) was used in the *in vitro* BBB/BTB permeability assays and had comparable effects to TY-52156 on TEER and doxorubicin permeability, and endothelial protein expression. Importantly, three other antagonists to various members of the S1P receptor family were also used, without effect. The panel of antagonists were used for both TEER and doxorubicin permeability readouts. These new data are in Fig. 5 of the revised manuscript.

4. Given the importance that the authors place on the inflammatory response, the reliance on immune-deficient models strongly limits interpretation of these results. If the authors wish to continue this line of reasoning, they will need to expand their models to include a syngeneic model. This will enable the use of a S1P3 knockout mouse as host.

Response: We thank the Reviewer for bringing up this pertinent point. A brain seeking variant of 4T1 syngeneic mouse model was added as an immunocompetent model of brain metastasis to ask whether higher BTB permeability was correlated with elevated S1P3 expression by neuro-inflammatory astrocytes. Similar to the three other brain metastasis models, an increase in astrocytic S1P3 was observed in the most permeable lesions. The data from this model was added to the revised Fig. 3, as well as to the revised Supplementary Figs. 3 and 4. These data bring to four the number of independent model systems demonstrating an association of astrocytic S1P3 overexpression and brain metastasis permeability. Unfortunately, knockout mice are performed in Bl/6 mice and the 4T1 model is Balb/c. There are no models of breast cancer brain metastases on a Bl/6 background.

Minor Points:

1. Immunofluorescence of normal brain and poorly permeable lesions should be presented alongside that from the highly permeable lesions in Figure 3A. This is described in the text but

not shown. This might be accomplished with a low-power image containing both the brain metastasis and surrounding adjacent normal brain parenchyma.

Response: This is a great suggestion. A low magnification picture was added to the revised Supplementary Fig. 2 to show that the uninvolved areas of the brain, i.e. further away from the metastatic lesions, do not have significant GFAP staining and do not have any detectable S1P3 staining.

2. The knock-down experiments (Figure 6) employ a single short hairpin to knock-down S1P3. To address the possibility of off target effects, these must be repeated with an additional independent short hairpin. Alternatively, CRISPRi might be employed for the same purpose.

Response: We thank the Reviewer for this comment. A set of experiments was performed with a new pair of shControl and shS1P3. Those experiments were added to the previous set of constructs in the revised Fig. 6. The two sets of shControl and shS1P3 presented similar patterns in terms of BBB and BTB permeability and endothelial junction protein expression.

3. Considerable work by other laboratories has taken place regarding astrocyte-brain metastatic cancer cell interactions. The authors should address the work by Kim SJ et al 2011 Neoplasia, as the datasets obtained after in vitro co-culture of MDA231 and astrocyte co-culture bear similarities to the data presented in this manuscript.

Astrocytes are likely key players in brain metastasis formation and evolution and multiple pathways have been identified. We have cited this reference in the Discussion section.

4. The manuscript is poorly prepared. Writing throughout the manuscript is informal, and filled with colloquialisms that are not readily understood by readers who do not speak English as a first language: e.g. line 60 “at levels higher than normal brain, but with peak levels ~a log less than”

Response: the symbol ~ was changed to “approximately”. The manuscript has been re-edited.

5. References are often absent or inadequate: e.g. line 70 “Interestingly, in two preclinical model systems, another 7% and 22-49% of metastases demonstrated >30 and >10 fold paclitaxel uptake about the BBB, respectively” (no reference given).

We deleted this sentence.

REVIEWERS' COMMENTS:

Reviewer #1 (Remarks to the Author):

The authors have addressed all of my points in a compelling way. I recommend acceptance of the manuscript.

Reviewer #2 (Remarks to the Author):

This improved manuscript addresses the clinically relevant problem of brain tumor barrier heterogeneity, the mechanism behind it and potential therapeutic targeting strategies to ameliorate it. Since last submission, the authors have greatly improved the clinical validation of the proposed pathway (namely signaling from the GPCR S1P3 in astrocytes). Of particular note, the authors took pains to expand the experimental repertoire to include a commercially-available antibody against S1P3 in FFPE tissue; allowing interested readers to expand the findings at their own institutions. They also present three S1P3 agonists in addition to the TY compound, lending credence to their observations, and lending credence to the assertion that this pathway represents a bona fide clinical target. Moreover, the authors have included an immune-competent syngeneic model, which enables more confident mechanistic conclusions. Finally, the revised manuscript is beautifully prepared, reads well and is readily understood by both native and non-native English speakers.